# Integrative genomics sheds light on the immunogenetics of tuberculosis in cattle
John F. O'Grady[1], Gillian P. McHugo [1], James A. Ward[1], Thomas J. Hall[1], Sarah L. Faherty O'Donnell[1,9], Carolina N. Correia [1,10], John A. Browne[1], Michael McDonald[1], Eamonn Gormley[2,3], Valentina Riggio [4,5], James G. D. Prendergast[4,5], Emily L. Clark [4,5], Hubert Pausch [6], Kieran G. Meade[1,3,7], Isobel C. Gormley[8], Stephen V. Gordon [2,3,7] & David E. MacHugh [1,3,7] ✉

*Mycobacterium bovis* causes bovine tuberculosis (bTB), an infectious disease of cattle that represents a zoonotic threat to humans. Research has shown that the peripheral blood (PB) transcriptome is perturbed during bTB disease but the genomic architecture underpinning this transcriptional response remains poorly understood. Here, we analyse PB transcriptomics data from 63 control and 60 confirmed *M. bovis*-infected animals and detect 2592 differently expressed genes perturbing multiple immune response pathways. Leveraging imputed genome-wide SNP data, we characterise thousands of *cis*-expression quantitative trait loci (eQTLs) and show that the PB transcriptome is substantially impacted by intrapopulation genomic variation during *M. bovis* infection. Integrating our *cis*-eQTL data with bTB susceptibility GWAS summary statistics, we perform a transcriptome-wide association study and identify 115 functionally relevant genes (including *RGS10*, *GBP4*, *TREML2*, and *RELT*) and provide important new omics data for understanding the host response to mycobacterial infections that cause tuberculosis in mammals.

Tuberculosis (TB) is a chronic infectious disease and a major source of ill health globally. Over one billion people have died as a consequence of human TB (hTB) during the past two centuries[1] and a further 1.25 million deaths were reported in 2023[2], illustrating both the historical and persistent threat of the disease. The primary causative agent of hTB, *Mycobacterium tuberculosis*, forms part of the *Mycobacterium tuberculosis* complex (MTBC), a group of phylogenetically closely related bacteria exhibiting extreme genomic homogeneity that causes TB disease in mammals[3–6]. Another member of the MTBC, *Mycobacterium bovis*, is the chief causative agent of bovine tuberculosis (bTB), an endemic disease principally associated with cattle that imposes a significant economic impact on individual farmers and national economies[7,8]. As a zoonotic pathogen, *M. bovis* can transmit from animals to humans causing zoonotic TB (zTB), which disproportionally affects the Global South[9,10]. The most recent estimates, available for 2019, attributed more than 140,000 new hTB cases and more than 11,000 deaths to zTB[11].

Previous research has shown that there are many shared characteristics between the pathogenesis of hTB and bTB, such that cattle can serve as a valuable large animal model to study TB disease in humans[12–15]. The primary route of infection for both *M. tuberculosis* and *M. bovis* is via the inhalation of aerosolised bacilli expelled by an infected individual or animal that are then phagocytosed by host alveolar macrophages (AMs), establishing the primary site of infection in the lung. Normally, efficient pathogen killing is achieved by AMs through a range of innate immune response mechanisms including encasement of the bacilli within a phagolysosome, autophagy and apoptosis of infected cells, and by the production of antimicrobial peptides[16,17]. However, tuberculous mycobacteria have evolved a range of strategies to manipulate innate immune responses, thereby facilitating colonisation, persistence, and replication within AMs[18–20]. Given the marked genomic similarities between *M. tuberculosis* and *M. bovis*, the close parallels between host-pathogen interactions and disease progression for hTB and bTB, and the zoonotic threat of *M. bovis*, a One Health approach to

[1]UCD School of Agriculture and Food Science, University College Dublin, Belfield, Ireland. [2]UCD School of Veterinary Medicine, University College Dublin, Belfield, Ireland. [3]UCD One Health Centre, University College Dublin, Belfield, Ireland. [4]The Roslin Institute and Royal (Dick) School of Veterinary Studies, University of Edinburgh, Midlothian, UK. [5]Centre for Tropical Livestock Genetics and Health (CTLGH), Roslin Institute, University of Edinburgh, Easter Bush Campus, Midlothian, UK. [6]Animal Genomics, ETH Zurich, Universitaetstrasse 2, Zurich, Switzerland. [7]UCD Conway Institute of Biomolecular and Biomedical Research, University College Dublin, Belfield, Ireland. [8]UCD School of Mathematics and Statistics, University College Dublin, Belfield, Ireland. [9]Present address: Irish Blood Transfusion Service, National Blood Centre, James's Street, Dublin, Ireland. [10]Present address: Children's Health Ireland, 32 James's Walk, Rialto, Ireland. ✉e-mail: david.machugh@ucd.ie

understanding the molecular mechanisms that underpin host immune responses and pathology in bTB can also provide important new information for tackling both hTB and zTB.

The genetic basis of susceptibility to *M. bovis* infection and bTB disease traits has been examined in cattle using focused candidate gene approaches[21–24]. Previous work has also highlighted the existence of substantial genetic variation for susceptibility to *M. bovis* infection in cattle populations[25,26]. In addition, genome-wide association studies (GWAS) have suggested susceptibility to *M. bovis* infection and bTB disease resilience traits are highly polygenic and influenced by interbreed genetic variation, which is reflected in modest replication of GWAS signals across multiple experiments[27–33]. In humans, genetic variation at proinflammatory gene loci contributes to resistance to *M. tuberculosis* infection[34]; an observation that has been supported by transcriptomic analysis of phenotypically defined resistant individuals[35]. Ultimately, for cattle, identifying, cataloguing, and measuring the functional effects of polymorphisms will expand and enhance genomic prediction models for economically important traits such as resistance to *M. bovis* infection[36].

Expression quantitative trait loci (eQTLs) are genomic sequence variations—primarily single-nucleotide polymorphisms (SNPs)—that modulate gene expression and mRNA transcript abundance[37–41]. In this regard, SNPs that are significantly associated with a trait of interest often exert an eQTL regulatory effect[42–44]. This is observed for hTB, where infection response eQTLs detected in dendritic cells challenged with *M. tuberculosis* were enriched for SNPs associated with susceptibility to hTB[45]. In cattle, eQTLs and other regulatory polymorphisms have been shown to contribute a substantial proportion of the genetic variation associated with multiple complex traits[46,47]. A transcriptome-wide association study (TWAS) is a multi-omics integrative strategy that combines gene expression data and independently generated GWAS summary data to discern explanatory links between genotypic variation, molecular phenotype variation, and phenotypic variation for a particular complex trait[48–52]. In this regard, the TWAS approach can provide meaningful insights into the molecular basis of quantitative trait loci (QTLs), and an integrated knowledgebase of tissue-specific human TWAS associations, the TWAS Atlas, has recently been developed[53]. TWAS approaches have also been leveraged to identify genes with expression patterns that modulate phenotypic variability for economically important traits in cattle[54,55]. Various TWAS methods have been developed to study the effects of proximal genetic variants (*cis*-eQTLs) on transcriptional regulation[48,49,56,57] that do not consider distal/interchromosomal regulatory polymorphisms (*trans*-eQTLs), which are a major component of the omnigenic model of complex trait inheritance[58]. To address this, the Multi-Omic Strategies for TWAS (MOSTWAS) suite of tools has been developed, which extend traditional TWAS approaches to include distal variants around regulatory biomarkers (e.g. transcription factor and microRNA genes) to increase the power to detect significant gene-trait associations[59].

It has previously been reported that the peripheral blood (PB) immune responses reflect those at the site of infection for bTB disease[60]. In this regard, our group and others have detected and characterised PB transcriptional biosignatures of *M. bovis* infection and bTB disease in cattle[61–71]. However, functional integration of PB transcriptomes, host genomic variation, and GWAS data sets for bTB susceptibility has not been performed previously. Additionally, to date, there have been no published studies that use the TWAS approach to understand the regulatory genome in the context of the host response to mycobacterial infections that cause TB in mammals. Therefore, using PB RNA-seq data from *M. bovis*-infected and control non-infected cattle, and imputed genome-wide SNP data, we conducted: (1) a high-resolution differential expression and functional enrichment analysis; (2) a genome-wide *cis*-eQTL analysis; and (3) a summary TWAS incorporating multiple bTB susceptibility GWAS data sets[33], which identified important new genes and variants underpinning the mammalian host response to mycobacterial infections that cause TB.

## Results

### Animal disease phenotyping

Figure 1 provides an overview of the experimental workflow and computational pipeline used for this study. A large panel of bTB reactor (bTB+; $n = 60$) and control (bTB−; $n = 63$) cattle were recruited that had a positive (reactor) and negative reaction, respectively, to the single intradermal comparative tuberculin test (SICTT). All animals were male, and the mean age of the animals was $21.9 \pm 8.3$ months. The Supplementary Table S1 provides detailed information about these animals, including the last four digits of the ear tag ID, date of sampling, and breed ancestry based on comprehensive pedigree information.

The mean $\Delta PPD$ ($\pm SE$) for the bTB+ animal group was $1170.35 \pm 84.48$ compared to $-360.46 \pm 55.17$ for the bTB− group and this group difference was highly significant (two-tailed Wilcoxon rank-sum test; $P < 3.258 \times 10^{-21}$) (Supplementary Fig. S1 and Supplementary Table S2). One designated bTB− control animal produced a positive result for the IFN-γ test (C050, $\Delta PPD = 263.1$) and two designated bTB+ animals elicited a negative result (T007, $\Delta PPD = 36.0$; T062, $\Delta PPD = -52.3$). These results yielded test sensitivity and specificity rates of 96.67% and 98.41%, respectively, which is in line with IFN-γ test performance under Irish conditions[72]. Based on their original SICTT results, these animals were still designated as bTB− and bTB+, respectively, and included in subsequent analyses.

### RNA-seq mapping statistics and genome-wide SNP imputation

Peripheral blood RNA sequencing yielded a mean of $35,129,315 \pm 3,430,729$ reads per individual sample library ($n = 123$ libraries and $\pm$ standard deviation). Reads were aligned to the ARS-UCD1.2 *Bos taurus* genome build with a mean of $33,352,903 \pm 3,206,593$ ($94.96 \pm 0.76\%$) reads mapping uniquely, $779,866 \pm 110,837$ ($2.22 \pm 0.17\%$) mapping to multiple loci, $14,168 \pm 2639$ ($0.04 \pm 0.008\%$) mapping to an excessive number of loci, $967,358 \pm 274,880$ ($2.74 \pm 0.68\%$) that were too short, and $15,020 \pm 2867$ ($0.04 \pm 0.008\%$) that could not be assigned to any genomic locus. The mean mapped length was $297.8 \pm 0.3$ bp (Supplementary Table S3). None of the libraries exhibited an abnormal distribution of gene counts (Supplementary Fig. S2).

A total of 591,947 array-genotyped SNPs were available for analysis. To determine if any animal samples were inadvertently duplicated, we first LD-pruned the array genotype data following filtering of variants that were rare (minor allele frequency (MAF) < 0.1) and that deviated from Hardy–Weinberg equilibrium (HWE; $P < 1 \times 10^{-6}$) to yield 34,272 SNPs. We then calculated the identity by state (IBS) among all animals using PLINK. We set a cut-off of 0.85 for deeming two samples as duplicates. All pairs of animals returned an IBS distance value < 0.8 (Supplementary Fig. S3a). Following this, we remapped the raw SNP array data from the UMD3.1 genome build to the ARS-UCD1.2 reference genome and imputed the remapped variants up to whole-genome sequence (WGS) scale using a Global Reference Panel as a reference[73] (Supplementary Note 2). Imputation performance improved as MAF increased with poor performance observed at variants with an MAF ≤ 1% (Supplementary Fig. S5). Following the removal of imputed variants that displayed poor imputation performance ($R^2 < 0.6$), possessed a low MAF (<0.05), and that deviated significantly from HWE ($P < 1 \times 10^{-6}$), a total of 3,711,843; 3,776,115; and 3,866,506 imputed autosomal SNPs were retained for the eQTL analysis in the control cohort (bTB−), reactor cohort (bTB+), and a combined all animals group (AAG), respectively. Lastly, a comparison of the imputed SNP profiles with RNA-seq reads using QTLtools showed that there were no sample mismatches and that the imputed WGS data correctly matched the transcriptomics data for all animals (Supplementary Fig. S3b).

### Population genomics, differential gene expression, and functional enrichment analyses

The results of the genetic structure analysis using the ADMIXTURE programme with 34,272 pruned genome-wide array SNPs and an inferred number of ancestral populations $K = 2$ are shown in Fig. 2a. A principal

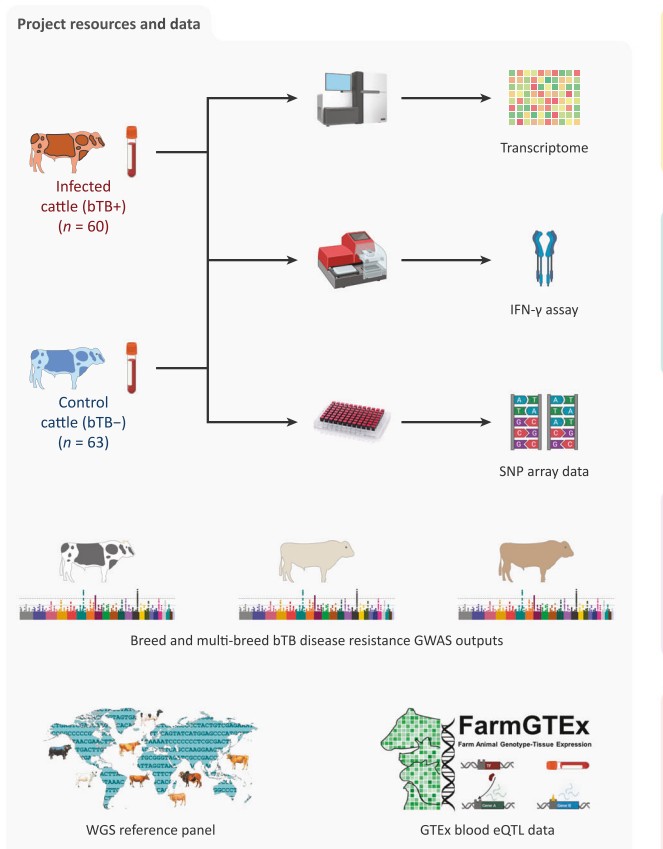

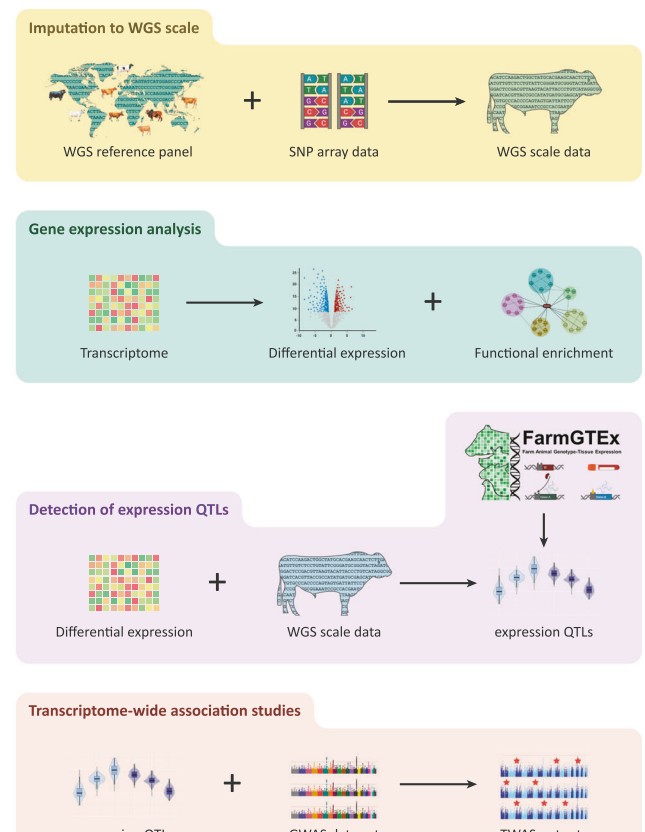

**Fig. 1 | Experimental and computational workflow.** Data resources for the project included; (1) newly generated high-resolution SNP-array data, peripheral blood RNA-seq data and interferon-gamma (IFN-γ) release assay (IGRA) measurements from a reference panel of $n = 60$ bovine tuberculosis (bTB) reactor (bTB+) and $n = 63$ control (bTB−) cattle; (2) single and multi-breed GWAS summary statistics for bTB susceptibility from Ring et al.,[33]; (3) whole genome sequence (WGS) data from Dutta et al., (2020)[73] and; (4) whole blood eQTL summary statistics from the Cattle Genotype-Tissue Expression (GTEx) Consortium[54]. For the reference panel, SNP array genotype data was remapped to the ARS-UCD1.2 bovine genome build and imputed using the WGS cohort as a reference panel. RNA-seq data was aligned to ARS-UCD1.2 with the resulting count matrices normalised using various

methodologies (see "Methods") for inclusion in the differential expression, functional enrichment, and expression quantitative trait loci (eQTL) analyses. The normalised expression matrix was integrated with the imputed SNP-array data for the eQTL analysis. To assess the replication of eQTLs, we leveraged whole-blood eQTL summary statistics from the Cattle GTEx Consortium[54]. Finally, the GWAS summary statistics were remapped to ARS-UCD1.2 before being integrated with the reference panel eQTL results to conduct three single- and one multi-breed transcriptome-wide association study (TWAS) for bTB susceptibility using the MOSTWAS software[59]. Some figure components were created with a BioRender. com license. Individual cattle art images are modified from Felius[162] with the permission of the author.

component analysis (PCA) plot of the first two principal components (PC1 and PC2) generated from the same set of pruned SNPs is shown in Fig. 2b with percentage Holstein ancestry and component 1 from the ADMIX-TURE structure analysis also shown for each animal sample (see also Supplementary Table S4). The results of these two analyses were in agreement: component 1 from the ADMIXTURE structure plot was in concordance with PC1 (10.8% of the variance derived from the top 20 PCs from the PCA), and likely corresponded, at least in part, to Holstein ancestry for the animals that had pedigree-derived breed composition data (112 out of 123 animals) (Fig. 2b and Supplementary Table S1). There was also a highly significant positive correlation (Spearman correlation ($\rho$) = 0.796; $P < 2.2 \times 10^{-16}$) between the pedigree-derived percentage Holstein ancestry values and component 1 from the ADMIXTURE structure plot (Supplementary Fig. S6a). PC2 (8.0% of the total variance of the top 20 PCs), on the other hand, likely accounts for population structure within the Holstein-Friesian populations, which has been documented previously in an independent cohort[27]. We observed that the genetic structure of the study population (bTB− and bTB+) was a confounder in the transcriptomics data set because there was sample clustering caused by breed ancestry observed in the PCA of the top 1500 most variable genes determined from the variance stabilised transformed count matrix in DESeq2 (Supplementary Fig. S6b).

We performed a differential expression analysis (DEA) to identify differentially expressed genes (DEGs) between the reactor (bTB+) and control (bTB−) animal groups, which incorporated genotype PC1 and PC2 (Fig. 2b), age in months, and sequencing batch (1 or 2) as covariates in the generalised linear model. With this approach, we identified 2592 DEGs (FDR $P_{adj.} < 0.05$) for the bTB+ vs bTB− contrast (Fig. 2c and Supplementary Table S5). Within the bTB+ group, increased expression was observed for 1638 DEGs and 954 DEGs exhibited decreased expression. We then selected a subset of 1091 highly significant DEGs (FDR $P_{adj.} < 0.01$) for gene set overrepresentation and functional enrichment analyses using g:Profiler and IPA®. In this subset of DEGs, 792 and 299 genes exhibited increased and decreased expression, respectively, in the bTB+ cohort.

Using the g:Profiler tool (FDR $P_{adj.} < 0.05$) we observed a clear enrichment for innate immune response, pathogen internalisation, and host-pathogen interaction Gene Ontology (GO) terms and biological pathways (Fig. 2d). The top significantly enriched functional entity was the *Defence response to virus* (FDR $P_{adj.} = 9.02 \times 10^{-13}$) GO Biological Process (GO:BP) term. Other significantly enriched functional entities included: *Cytosolic pattern recognition receptor signalling pathway* (FDR $P_{adj.} = 1.24 \times 10^{-4}$) GO:BP term; *RIG-I-like receptor signalling pathway* (FDR $P_{adj.} = 3.88 \times 10^{-5}$) from the Kyoto Encyclopaedia of Genes and Genomes; and *Antiviral mechanism by IFN-stimulated genes*

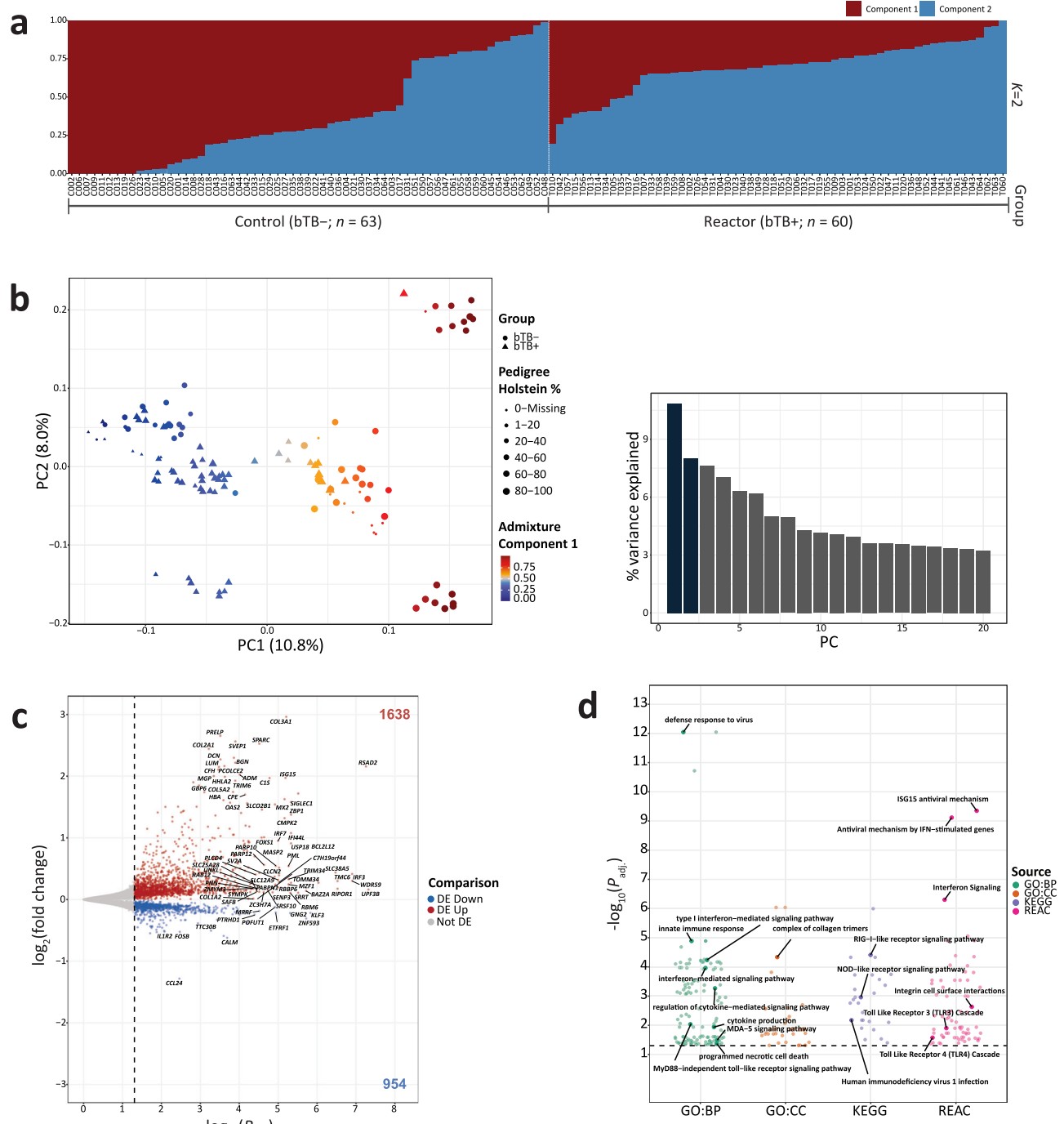

**Fig. 2 | Population genomics, differential expression, and functional enrichment analyses. a** Structure plot showing the proportion of ancestry components 1 and 2 from the ADMIXTURE analysis for 123 animals (reactor bTB+ and control bTB−). **b** Principal component analysis (PCA) plot of PC1 and PC2 derived from 34,272 pruned SNPs genotyped in 123 animals. The data points are shaped based on their experimental designation, coloured based on the inferred ancestry component 1 from the ADMIXTURE analysis, and sized based on their reported pedigree Holstein percentage. A histogram plot of the relative variance contributions for the first 20 PCs is also shown with PC1 and PC2 accounting for 10.8% and 8.0% of the variation in the top 20 PCs, respectively. **c** Horizontal volcano plot of differentially expressed genes (DEGs) for the bTB+ ($n = 60$) vs bTB− ($n = 63$) contrast with thresholds determined by FDR $P_{adj.} < 0.05$ and an absolute $\log_2$ fold-change (LFC) > 0. The $x$-axis represents the $-\log_{10} P_{adj.}$ and the $y$-axis represents the LFC. **d** Jitter plot of significantly impacted pathways/GO terms identified across the gene ontology (GO) biological processes (GO:BP), cellular component (GO:CC), reactome (REAC) and Kyoto Encyclopaedia of Genes and Genomes (KEGG) databases using g:Profiler. The data points are coloured according to the corresponding database.

(FDR $P_{adj.} = 7.67 \times 10^{-10}$) from the Reactome database. All significant results obtained from g:Profiler, in addition to the intersection of DEGs with the respective functional entities, are provided in Supplementary Table S6. For IPA®, a total of 996 DEGs and 14,228 background genes were successfully mapped. The significantly enriched (FDR $P_{adj.} < 0.05$) pathways

identified from IPA® included *Interferon alpha/beta signalling* (FDR $P_{adj.} = 4.90 \times 10^{-8}$), *Oxidative phosphorylation* (FDR $P_{adj.} = 2.51 \times 10^{-6}$) and *Activation of IRF by cytosolic pattern recognition receptors* (FDR $P_{adj.} = 9.12 \times 10^{-3}$) (Supplementary Fig. S7a and Supplementary Table S7). Upstream transcriptional regulator analysis using IPA® revealed that the

transcriptional regulator, ETV3 was the most significant biological regulator of the inputted DEGs (FDR $P_{adj.}$ = 4.89 × 10$^{-19}$) (Supplementary Fig. S7b). Other important statistically significant (FDR $P_{adj.}$ < 0.05) upstream regulators include TLR3, STING1, IRF5, and STAT1 (for complete results see Supplementary Table S8).

### Mapping *cis*-expression quantitative trait loci and estimating the contribution of non-additive effects

We used a linear regression model in TensorQTL to test associations between expressed genes and SNPs that passed filtering thresholds to identify local (±1 Mb) *cis*-eQTLs in the reactor (bTB+) group (*n* = 60), the control (bTB−) group (*n* = 63), and a combined all animals group (AAG, *n* = 123). As covariates, we also included: (1) the top five SNP genetic variation PCs (PC1-5) that accounted for between 37.8-41.7% of the variation in the top 20 genotype PCs inferred for each group separately; (2) age in months; (3) sequencing batch; (4) disease status (where applicable); and (5) transcriptomic PCs with PC1-8, PC1-9, and PC1-14 included for the bTB+, bTB−, and AAG cohorts, respectively. The number of transcriptomic PCs to use was determined using the elbow method (Supplementary Fig. S8). We also removed known covariates (genotype PCs, age, batch, disease status) that were well captured by the inferred covariates (unadjusted $R^2$ ≥ 0.9) and the final set of covariates for each cohort are detailed in Supplementary Tables S9-11. In total, we tested 14,701; 14,598; and 14,612 genes in the bTB+, bTB−, and AAG cohorts, respectively, for *cis* SNP variants associated with their expression levels.

Table 1 summarises the number of significant (FDR $P_{adj.}$ < 0.05) *cis*-eQTLs, *cis*-eVariants, and *cis*-eGenes identified in all three groups. We identified 2219, 3366, and 6676 *cis*-eGenes in the bTB+, bTB−, and AAG cohorts, respectively, with the largest proportion captured by the AAG cohort (Fig. 3a and Supplementary Tables S12–14). For each *cis*-eGene in each group, variants with a nominal *P*-value below the gene-level threshold (Supplementary Fig. S9) were considered significant *cis*-eVariants. Overall, we identified 167,353; 410,867; and 1,103,004 significant *cis*-eVariant:gene associations in the bTB+, bTB−, and AAG cohorts, respectively. Of these *cis*-eVariants, 17.1%, 23.4% and 35.7% were associated with >1 *cis*-eGene, respectively. For all three groups, we identified hundreds to thousands of *cis*-eGenes with multiple independent acting *cis*-eQTLs (Fig. 3b). The conditional analysis detected 12.5%, 25.9%, and 82.2% additional independent *cis*-eQTLs in the bTB+, bTB−, and AAG cohorts, respectively. We observed that the top *cis*-eQTL identified by the permutation analysis tended to cluster close to the transcriptional start site (TSS) of the associated gene, whereas conditionally independent *cis*-eQTLs were located at variable distances to the TSS (Wilcoxon rank-sum test; $P < 2.2 \times 10^{-16}$) (Fig. 3c). The permuted and conditional *cis*-eQTL associations were symmetrical around the TSS with no enrichment in the 5' or 3' directions (Supplementary Fig. S10). We noted a moderately negative but highly significant Spearman correlation ($\rho$) in the effect size estimates of *cis*-eQTLs and their respective distances to the TSS of the associated gene in all three groups (Fig. 3d).

**Table 1 | Total number of and unique number of significant (FDR $P_{adj.}$ < 0.05) *cis* expression quantitative trait loci (*cis*-eQTLs), *cis*-eVariants and their corresponding *cis*-eGenes identified across the reactor (bTB+), control (bTB−), and combined all animals group (AAG) cohorts, respectively**

| Class of eQTL/eGene | bTB + (*n* = 60) | bTB − (*n* = 63) | AAG (*n* = 123) |
|---|---|---|---|
| *cis*-eQTLs (permutation) | 2219 | 3366 | 6676 |
| Conditionally independent *cis*-eQTLs | 278 | 872 | 5489 |
| Total number of independent *cis*-eQTLs | 2497 | 4238 | 12,165 |
| *cis*-eGenes | 2219 | 3366 | 6676 |
| *cis*-eVariant associations | 167,353 | 410,867 | 1,103,004 |
| Unique *cis*-eVariants | 138,677 | 314,891 | 709,337 |

To assess replication of the *cis*-eQTLs identified in this study, we used three metrics: allelic concordance (AC), the $\pi_1$ statistic to measure the proportion of true positive associations, and the Spearman correlation ($\rho$) coefficient of effect size estimates in an external set of whole-blood *cis*-eQTL summary statistics obtained from the Cattle Genotype Tissue Expression (GTEx) Consortium[54]. We observed high AC between the top and significant *cis*-eQTLs identified in this study and significant *cis*-eQTLs identified in the Cattle GTEx Consortium (AC$_{bTB+}$ = 99.27%, AC$_{bTB-}$ = 99.10%, and AC$_{AAG}$ = 98.87%). We observed moderate to high $\pi_1$ statistics (± SEM) across all groups indicating good replication ($\pi_{1bTB+}$ = 0.794 ± 0.004, $\pi_{1bTB-}$ = 0.70 ± 0.003, and $\pi_{1AAG}$ = 0.611 ± 0.003) (Fig. 3e). We also noted a positive and significant Spearman correlation ($\rho$) in effect size estimates for the top significant eQTLs identified in our study and the matched variants from the Cattle GTEx Consortium across all three groups ($\rho_{bTB+}$ = 0.797, $\rho_{bTB-}$ = 0.786, and $\rho_{AAG}$ = 0.761) (Fig. 3f).

Across all genes tested in the *cis*-eQTL mapping analysis for the AAG cohort, we constructed an additive genomic relationship matrix (GRM) for *n* = 14,563 of these genes and observed that the median *cis* narrow-sense heritability of gene expression (*cis*-$h^2_{SNP}$) was 0.165 (Supplementary Fig. S11a). We noted a significant difference in the median *cis*-$h^2_{SNP}$ estimates of genes with expression levels considered significantly heritable (Likelihood ratio test (LRT) $P_{adj.}$ < 0.05) (*n* = 6757; median *cis*-$h^2_{SNP}$ = 0.413) to those that were not significantly heritable (*n* = 7806; median *cis*-$h^2_{SNP}$ = 0.04) (Wilcoxon rank-sum test; $P < 2.2 \times 10^{-16}$) (Supplementary Fig. S11b). For the 6757 genes with significantly heritable expression levels, we were able to construct dominance GRMs for 5863 (86.7%) of these genes. For these genes, we estimated the median *cis* broad-sense heritability ($H^2_{SNP}$) of gene expression to be 0.474, the median *cis*-$h^2_{SNP}$ to be 0.431 and the median *cis* heritability of dominance effects (*cis*-$\delta^2_{SNP}$) to be 6 × 10$^{-6}$ with the difference between the two latter categories being statistically significant (Wilcoxon rank-sum test; $P < 2.2 \times 10^{-16}$) (Supplementary Fig. S11c). Out of the 5863 genes tested in this analysis, we observed two genes (0.03%) for which the dominance component contributed a significant proportion to the *cis*-$H^2_{SNP}$ of gene expression. These two genes were *LANCL1* (LRT $P_{adj.}$ = 0.01; *cis*-$h^2_{SNP}$ = 0.314; *cis*-$\delta^2_{SNP}$ = 0.489) and the unannotated *ENSBTAG00000031825* gene (LRT $P_{adj.}$ = 0.03; *cis*-$h^2_{SNP}$ = 0.914; *cis*-$\delta^2_{SNP}$ = 0.057).

We also tested to infer what proportion of *cis*-eQTLs mapped using the linear model could be better explained by the addition of a non-additive (dominance) component. Of the 2497, 4238 and 12,165 independent *cis*-eQTLs identified in the bTB+, bTB−, and AAG cohorts, respectively, we fitted a non-additive component in the linear model for 2236, 3737 and 11,092 *cis*-eQTLs, as these *cis*-eQTLs had ≥1 animal in both homozygous groups. Of these *cis*-eQTLs tested for evidence of non-additive effects in the bTB+, bTB− and AAG cohorts, we identified a total of three, four and 31 independent *cis*-eQTLs displaying significant evidence of a deviation from additivity (LRT $P_{adj.}$ < 0.05), which equated to 0.13%, 0.11%, and 0.28% of all tested *cis*-eQTLs in each group, respectively. Overall, these results indicate that dominance variance makes a negligible contribution to the *cis*-heritability of gene expression and mapping of *cis*-eQTLs in cattle is robustly supported by an additive model.

### Peripheral blood *cis*-eVariants have context-specific effects during *M. bovis* infection and bTB disease

The beta coefficients from the linear model in TensorQTL of top *cis*-eQTLs were quantified in normalised space; therefore, the biological effect of top *cis*-eQTLs in each group was quantified as the log$_2$ allelic fold-change (slope_aFC) in the expression of the associated *cis*-eGene[74]. As expected, we observed that as sample size increased, smaller eQTL effects were observed with the median |slope_aFC| of *cis*-eQTLs being equal to 0.30 in the AAG cohort, 0.42 in the bTB− cohort, and 0.54 in the bTB+ cohort, respectively. In addition, these slope value distributions were significantly different from each other (Wilcoxon rank-sum test; $P < 2.2 \times 10^{-16}$) (Fig. 4a). We also noted a strongly positive and significant correlation (Spearman correlation ($\rho$) ≥ 0.90; $P < 2.2 \times 10^{-16}$) between the slope_aFC values and the beta

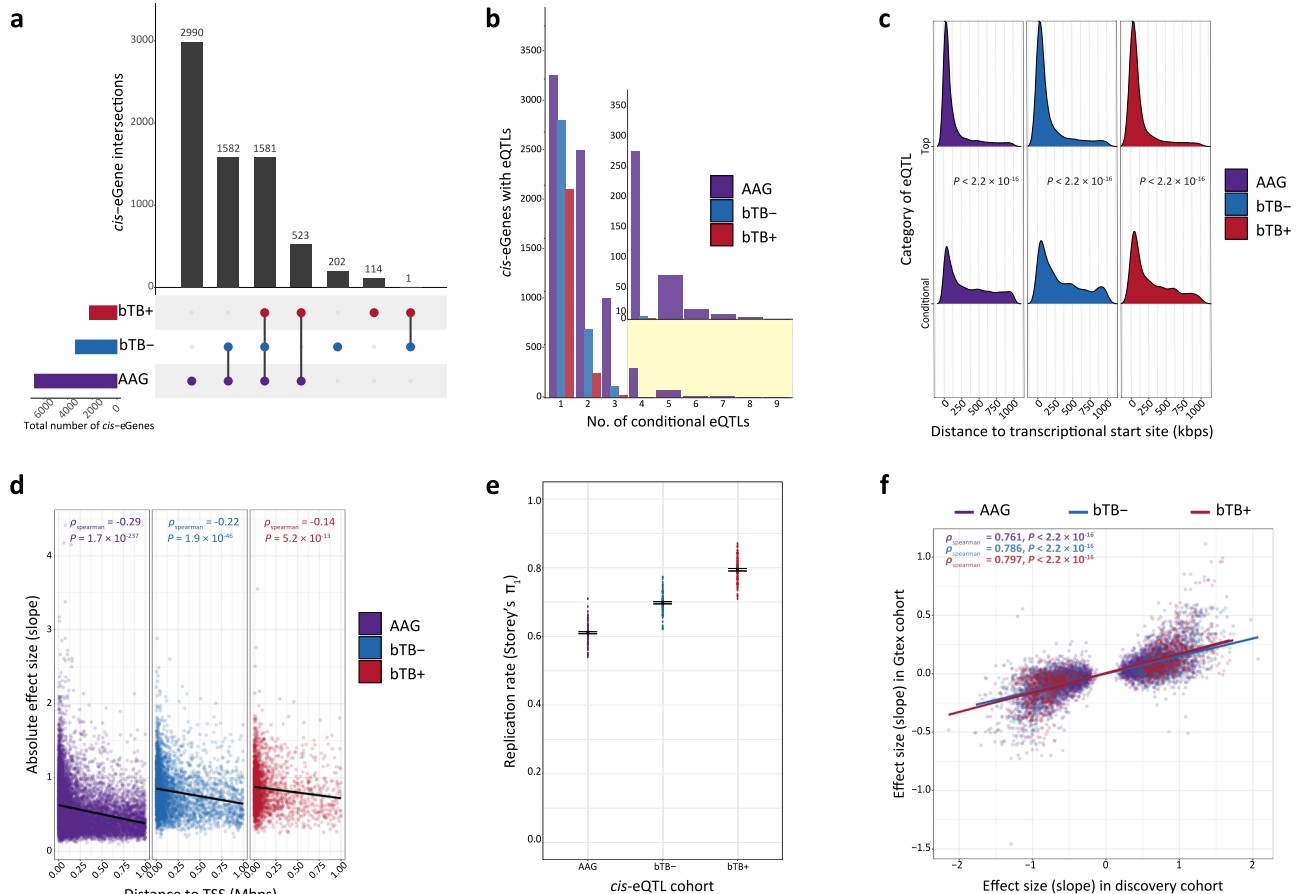

**Fig. 3 | Cis-expression quantitative trait loci (cis-eQTL) mapping and external replication results. a** Upset plot showing the intersection of shared *cis*-eGenes identified in the reactor (bTB+), control (bTB−), and combined all animals group (AAG) cohorts, respectively. **b** Barplot illustrating the number of genes with a significant primary or conditional *cis*-eQTL for degrees 1–9 across all three groups. The inset shows the number of genes for *cis*-eQTL degrees 4–9. **c** Ridgeline plot showing the distribution of the absolute distance from the transcriptional start site (TSS) of top and conditional *cis*-eQTLs identified in all three groups. *P*-values are inferred from the Wilcoxon rank-sum test between top and conditional *cis*-eQTLs within each group. **d** Scatter plot illustrating the relationship between absolute *cis*-eQTL effect size and distance to the TSS for all significant *cis*-eQTLs (top and conditional) identified in each group separately. Spearman correlation (ρ) values are

also reported in addition to the corresponding *P*-value representing the significance level of each respective correlation. The black line indicates line of best fit. **e** Replication rate as measured by Storey's $\pi_1$ in the current study and whole blood *cis*-eQTLs identified in the Cattle GTEx[54]. The error bars indicate the standard error of the sample mean (SEM) calculated from 100 bootstrap samplings. **f** Scatterplot illustrating the effect sizes of significant *cis*-eQTLs identified in this study and matched variant-gene pairs identified in the Cattle GTEx. Spearman correlation values are also reported in addition to the corresponding *P*-value representing the significance level of each respective correlation. The coloured lines indicate lines of best fit within each group, respectively. The colour scheme for each group is consistent throughout the figure.

coefficients from the linear model used for eQTL mapping with TensorQTL (Fig. 4b). Across the three groups, 1098 of the top *cis*-eQTLs (16.4%) had large effects (defined as |slope_aFC| ≥ 1) in the AAG cohort, 799 eQTLs (23.7%) had large effects in the bTB− group, and 669 (30.1%) had large effects in the bTB+ group. Notably, and consistent with previous studies[75], we also observed a pattern in the relationship between eQTL effect size and MAF distribution, whereby large-effect *cis*-eQTLs tended to have low MAFs and this pattern was more pronounced in the bTB− and bTB+ groups in comparison to the AAG cohort (Supplementary Fig. S12).

In this experiment, a traditional and paired response-eQTL analysis, which identifies eQTLs in the same set of subjects before or after the application of a stimulus/treatment but not in both, or that characterises eQTLs with opposing effects in both groups was not possible as different animals were present in the bTB− and bTB+ cohorts. However, to identify genetic variation that is associated with different effects on gene expression levels in the bTB+ and bTB− groups, we conducted an interaction analysis and fitted an interaction term in the AAG cohort between the genotype at a locus and bTB status ($g{\times}i$) in the TensorQTL linear model accounting for the same covariates used in the initial *cis*-eQTL mapping procedure. From this analysis, we identified a total of five *cis*-eQTLs significantly (FDR

$P_{adj.} < 0.25$) interacting with bTB status (Table 2). The genes associated with these interacting *cis*-eQTLs (ieQTLs) included *PCBP2* (Fig. 4c)*, PLD3, RELT, GBP4* and *PNPLA1*. We note that previous studies have used this relaxed FDR $P_{adj.}$ cut-off, illustrating that identifying ieQTLs interacting with higher-order traits is challenging, but that failure to detect an ieQTL effect does not indicate the absence of an ieQTL effect[76,77]. Therefore, to identify putative context-specific *cis*-eVariants, we compared the effect of top *cis*-eQTLs for common genes and SNPs tested in both the bTB− and the bTB+ groups, because substantial differences in these effects may reflect or indicate context-specific regulatory mechanisms. To do this, for each gene, we selected the variant:gene pair with the lowest FDR $P_{adj.}$ value in either group (i.e. the most significant eQTL) and obtained the corresponding eQTL summary statistics for this variant from both groups. We then determined whether this extracted variant was a *cis*-eVariant in both groups (i.e. if the nominal *P*-value of association was less than the gene-level threshold (Supplementary Fig. S9)). If the variant was a *cis*-eVariant in the bTB− group but not the bTB+ group, it was considered a bTB− magnified *cis*-eVariant. Conversely, if it was a *cis*-eVariant in the bTB+ cohort but not the bTB− cohort, it was considered a bTB+ magnified *cis*-eVariant. If it was a *cis*-eVariant in both groups, it was considered a shared *cis*-eVariant. If the

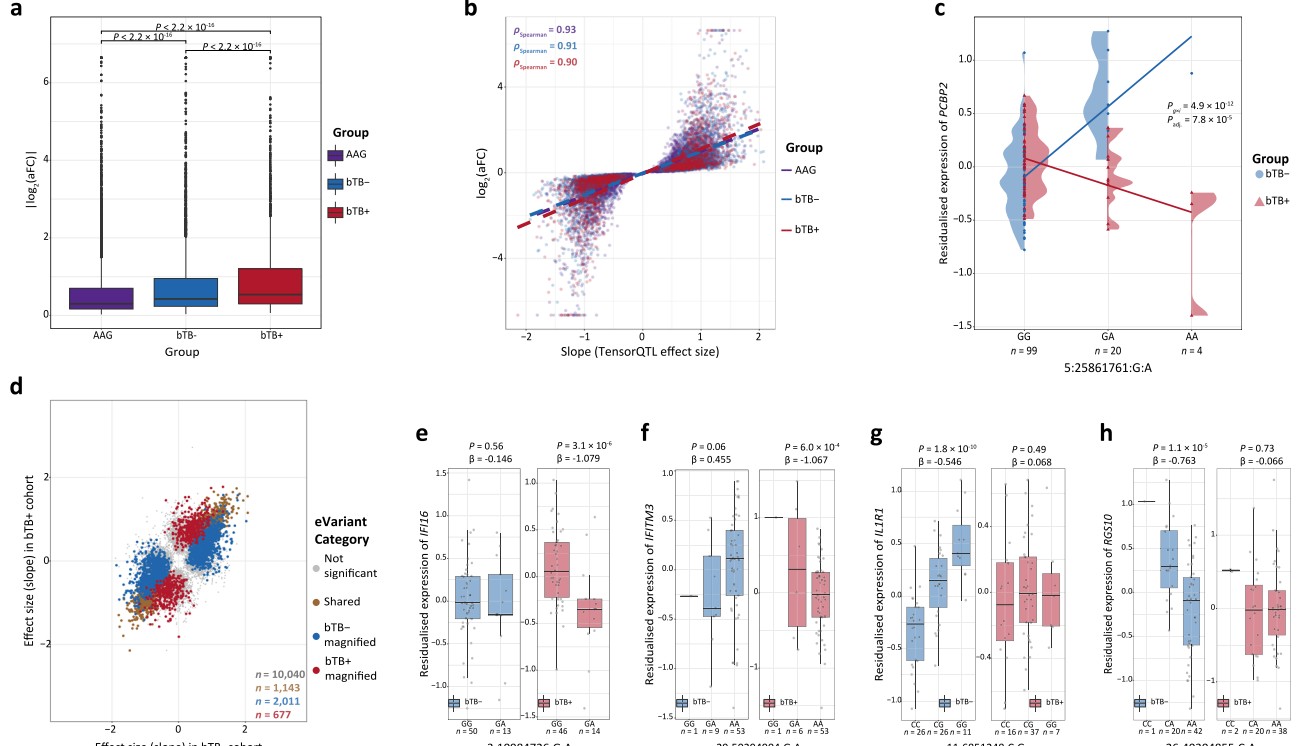

**Fig. 4 | Effect of sample size on *cis*-QTL discovery and the characterisation of interaction *cis*-eQTLs and context-specific *cis*-eVariants. a** Boxplot showing the distribution of the absolute $\log_2$ allelic fold change (slope_aFC) for top significant *cis*-eQTLs ($P_{adj.} < 0.05$) identified in the all animals group (AAG), the control group (bTB−) and the reactor group (bTB+), respectively. **b** Scatter plot showing the Spearman correlation ($\rho$) between the slope_aFC and the effect size from the linear regression model of top significant *cis*-eQTLs in the AAG, bTB−, and bTB+ cohorts, respectively. **c** Visualisation of an interaction *cis*-eQTL (ieQTL) for the gene *PCBP2* identified in the AAG cohort. Data points are coloured and shaped based on an animal's experimental designation. Trend lines are inferred from setting the *geom_smooth* function in R setting the method to "*lm*". **d** Scatter plot illustrating the effect sizes of *cis*-eVariants tested in both the bTB− and bTB+ cohorts. *Cis*-

eVariants are coloured depending on whether they are significant ($P_{adj.} < 0.05$) in the bTB− cohort only (blue), in the bTB+ cohort only (red), in both the bTB− and bTB+ cohorts (gold), or neither (grey). **e** Boxplot displaying the residualized expression values for the gene *IFI16*, with individuals separated based on their genotype at the SNP Chr3:10984726:G:A and coloured based on their respective cohort. Nominal *P*-values of association in each cohort are displayed with slope values from the linear regression model. **f** Same as (**e**) but for the gene *IFITM3* and the SNP Chr29:50294904:G:A. **g** Same as (**e**) but for the gene *IL1R1* and the SNP Chr11:6851248:C:G. **h** Same as (**e**) but for the gene *RGS10* and the SNP 26:40304855:C:A. The box plots in (**a**) and **e**–**h** cover the interquartile range with the median line denoted at the centre, and the whiskers extend to the most extreme data point that is no more than 1.5 × IQR from the edge of the box.

variant was not a *cis*-eVariant in either group, it was designated as non-significant. Using this approach, for 13,871 common gene-variant pairings, we characterised 1143 shared *cis*-eVariants, 2011 bTB− magnified *cis*-eVariants, 677 bTB+ magnified *cis*-eVariants and 10,040 non-significant *cis*-eVariants. (Fig. 4d; Supplementary Table S15). Notably, none of the shared *cis*-eVariants exhibited discordance in terms of effect size.

We did not observe a significant association between the genes that the bTB+ magnified *cis*-eVariants putatively regulate and genes considered significantly (FDR $P_{adj.} < 0.05$; Fig. 2c) differentially expressed between the bTB+ and bTB− groups (chi-square test; $\chi^2 = 0.02$; $P = 0.882$). Similarly, we did not identify a significant association between bTB− magnified *cis*-eGenes (chi-square test; $\chi^2 = 0.32$; $P = 0.5692$), nor for a combined group of bTB+ and bTB− magnified *cis*-eGenes (chi-square test; $\chi^2 = 0.38$; $P = 0.54$). These results suggest that the DEGs are not being inflated by different genetic structures in the bTB+ and bTB− groups (Fig. 2a), which is to be expected given that we controlled for such genetic structures in the DEA. However, we did identify variants associated with key interferon-inducible genes that have different magnitudes of effect on gene expression following *M. bovis* infection such as the bTB+ magnified *cis*-eVariant Chr3:10984726:G:A associated with decreased expression of *IFI16* ($P = 3.1 \times 10^{-6}$) (Fig. 4e), but that displayed increased expression in the bTB+ group based on the DEA (Supplementary Table S5). Similarly, we also identified the bTB+ *cis*-eVariant Chr29:50294904:G:A as being associated with decreased expression of *IFITM3* in the bTB+ cohort

**Table 2 | Significant (FDR $P_{adj.} < 0.25$) interaction *cis*-expression quantitative trait loci results for the interaction term of genotype and bTB status ($g{\times}i$) fitted in the all animals group (AAG) cohort using TensorQTL[141]**

| Variant | Gene | *P*-value$_{g{\times}i}$ | eigenMT $P_{adj.}$ | FDR $P_{adj.}$ |
|---|---|---|---|---|
| Chr5:25861761:G:A | *PCBP2* | $4.9 \times 10^{-12}$ | $5.3 \times 10^{-9}$ | $7.8 \times 10^{-5}$ |
| Chr18:49353483:G:C | *PLD3* | $8.0 \times 10^{-10}$ | $8.8 \times 10^{-7}$ | $6.4 \times 10^{-3}$ |
| Chr15:53330033:T:C | *RELT* | $1.3 \times 10^{-8}$ | $1.7 \times 10^{-5}$ | 0.08 |
| Chr3:53539162:G:T | *GBP4* | $5.4 \times 10^{-8}$ | $6.8 \times 10^{-5}$ | 0.22 |
| Chr23:10034540:C:T | *PNPLA1* | $6.9 \times 10^{-8}$ | $7.6 \times 10^{-5}$ | 0.22 |

The *cis* variant and corresponding *cis* gene are displayed, along with the nominal *P*-value of the interaction association, the gene-level eigenMT adjusted *P*-value[145] and the global false discovery rate (FDR) adjusted *P*-value.

($P = 6.0 \times 10^{-6}$), an effect that was divergent but not significant in the bTB− cohort ($\beta_{bTB−} = 0.455$, $\beta_{bTB+} = -1.067$) (Fig. 4f). With respect to bTB− magnified *cis*-eVariants, we identified the SNP Chr11:6851248:C:G as being significantly associated with increased expression of *IL1R1* ($P = 1.8 \times 10^{-10}$) (Fig. 4g), and the SNP Chr26:40304855:C:A associated with decreased expression of *RGS10* ($P = 1.1 \times 10^{-5}$) (Fig. 4h). Other relevant genes from this analysis included *CLEC4E*, *GBP4*, and *JAK2* that were associated with

bTB+ magnified cis-eVariants, and *CXCR4, IL18BP*, and *NLRC4* that were associated with the bTB− magnified cis-eVariants (Supplementary Table S15). Collectively, our analysis reveals that although we are limited to detecting cis-eVariants/cis-eQTLs with large effects, sample sizes of $n \geq 60$ in each group provide sufficient power to detect putative context-specific cis-eVariants associated with immunologically relevant genes in PB with differing effects between bTB− and bTB+ animals.

## Transcriptome-wide association analyses highlight genes associated with bTB susceptibility

To assess if expression patterns in the three groups of animals (bTB+, bTB−, and AAG) were correlated to bTB susceptibility, we used MOSTWAS[59] to generate predictive models of expression and combined these, using a TWAS approach, with SNP summary statistics from multiple GWAS data sets for bTB susceptibility in four breed cohorts (Holstein-Friesian—HF, Charolais—CH, Limousin—LM, and a multi-breed panel—MB)[33]. The SNPs in these GWAS data sets were originally analysed on the UMD3.1 genome assembly and were therefore remapped to the ARS-UCD1.2 assembly for this TWAS. We first computed 28,599, 88,525, and 1,046,632 significant (FDR $P_{adj.} < 0.01$) correlations between expressed transcription factors and co-transcription factors (TFs/co-TFs) that were considered cis-eGenes, and all other cis-eGenes in the bTB+, bTB−, and AAG cohorts, respectively. We then used the *MeTWAS* function in MOSTWAS to build predictive models of expression for cis-eGenes within each group. In total, we trained 1594, 2493, and 3954 expression models in the bTB+, bTB−, and AAG cohorts, respectively (Table 3). The expression patterns of these genes were significantly heritable ($P < 0.05$) and achieved McNemar's five-fold cross-validated predicted $R^2$ value $\geq 0.01$ within the *MeTWAS* function. For each reference group and each GWAS cohort, we conducted a weighted burden test using the MOSTWAS *BurdenTest* function to identify genes with expression patterns correlated to bTB susceptibility. For genes that were significant at a Bonferroni-adjusted *P*-value < 0.05, we conducted a permutation test conditioning on the GWAS effect size and genes with a permuted *P*-value < 0.05 were considered significantly associated with bTB susceptibility.

The number of genes that were significant after Bonferroni correction, and that remained significant after the permutation procedure in each of the 12 TWAS groups are shown in Table 3. In total, across all four GWAS cohorts (HF, CH, LM, and MB) we identified 30, 26, and 70 TWAS genes significantly associated with bTB susceptibility in the bTB+, bTB−, and AAG cohorts, respectively (Fig. 5). Among the cohorts, there was little overlap between TWAS genes, with many genes emerging as breed- and expression model-specific (Supplementary Fig. S13). Overall, we identified 115 genes dispersed across the genome with expression patterns correlated with bTB susceptibility (Fig. 5). Our TWAS analysis highlighted immunobiologically relevant genes such as the signalling lymphocyte-activation molecule family 1 gene (*SLAMF1*; $P = 3.3 \times 10^{-6}$; $Z = 4.7$) in the bTB+:MB TWAS cohort, the regulator of G-protein signalling 10 gene (*RGS10*; $P = 2.7 \times 10^{-19}$; $Z = 9.0$) in the bTB−:HF TWAS cohort, the guanylate binding protein 4 gene (*GBP4*; $P = 1.23 \times 10^{-8}$; $Z = -5.7$) in our AAG:CH TWAS cohort, the triggering receptor expressed on myeloid like cells 2 gene (*TREML2*; $P = 7.6 \times 10^{-9}$; $Z = 5.8$) in the bTB−:HF TWAS cohort, and the

RELT TNF receptor gene (*RELT*; $P = 4.4 \times 10^{-9}$; $Z = 5.9$) in our AAG:HF TWAS cohort with the full results of all TWAS associations for each reference panel provided in Supplementary Tables S16–18. For interpreting the effect of these significant genes, a negative *Z*-score indicates that decreased expression of a gene is associated with bTB susceptibility, whereas a positive *Z*-score indicates overexpression of a gene is associated with the trait of interest[49].

## Discussion

We present a comprehensive multi-omics analysis, which integrates genomics, bovine PB transcriptomics, and GWAS data sets for bTB susceptibility to improve our understanding of how genetic factors contribute to the intrapopulation variability in response to *M. bovis* infection and mycobacterial infections more broadly. Moreover, this study is the first application of the TWAS approach to dissecting the genomic architecture of a susceptibility trait for a mycobacterial infection that causes TB in mammals. Additionally, as our data set represents one of the few in vivo cis-eQTL mapping analyses conducted for cattle with an infectious disease[78], our eQTL and TWAS results align with and support the Cattle GTEx Consortium's objectives of: (1) mapping context-specific cis-eQTLs across different conditions; (2) contributing to the precision breeding of disease (in this case bTB) resistant animals; and (3) serving as a reference for veterinary medicine and comparative genomics, particularly in relation to hTB under a One Health framework.

Bovine TB disease susceptibility is a moderately heritable quantitative trait (estimated $h^2$ ranges between 0.08 and 0.14) with a highly polygenic and breed-specific genetic architecture that poses significant challenges for functional assignment of QTLs identified from GWAS experiments[33,79–81]. However, understanding the biology of these QTLs will be important in bridging the genome to phenome gap for bTB disease resistance because regulatory QTLs, especially cis-eQTLs, contribute a large proportion of the variance in complex trait heritability[46,47]. Additionally, it has been estimated that up to 50% of GWAS signals are shared with at least one molecular phenotype in humans[82], with a particular enrichment observed for regulatory QTLs associated with proximal and distal gene expression regulation in PB[83].

Analysis of differential gene expression using RNA-seq showed that the bovine PB transcriptome is substantially perturbed by *M. bovis* infection with 2592 significant DEGs (FDR $P_{adj.} < 0.05$) observed (Fig. 2c and Supplementary Table S5). Many of the DEGs detected here (38%) were also observed to be differentially expressed at 48 h post-infection (hpi) in bovine alveolar macrophages (bAMs) challenged with *M. bovis* and were components of gene modules key to the innate immune response[84]. These shared genes included, but were not limited to *MX2, MX1, OAS2, ISG15*, and *IRF7*, which collectively constitute interferon-stimulated genes[85], again indicating that peripheral immune responses reflect those at the site of mycobacterial infection in the lung[60,86]. This is also reflected in our gene set enrichment analyses of highly significant DEGs where many of the top significantly overrepresented functional entities were biological pathways and GO terms related to interferon signalling and induction of interferon genes (Fig. 2d and Supplementary Table S6), and is consistent with previous blood-based transcriptional studies of both hTB and bTB that suggest proinflammatory

**Table 3 | Total number of significantly heritable ($P < 0.05$) predictive expression models ($R^2 > 0.01$) generated for the reactor (bTB+), control (bTB−) and combined all animals group (AAG) cohorts with the corresponding Bonferroni adjusted *P*-value cut-off for association and number of significant genes identified across all four GWAS data sets**

| Group | Expression models | *P*-value cut-off | Limousin (LM) TWAS genes | Holstein-Friesian (HF) TWAS genes | Charolais (CH) TWAS genes | Multi-breed (MB) TWAS genes |
|---|---|---|---|---|---|---|
| bTB+ | 1594 | $3.14 \times 10^{-5}$ | 8 (3) | 13 (8) | 15 (7) | 28 (13) |
| bTB− | 2493 | $2.00 \times 10^{-5}$ | 8 (2) | 26 (12) | 12 (3) | 17 (10) |
| AAG | 3954 | $1.26 \times 10^{-5}$ | 22 (13) | 46 (32) | 24 (10) | 53 (19) |

Numbers in brackets indicate the number of transcriptome-wide association study (TWAS) genes significant after permutation testing

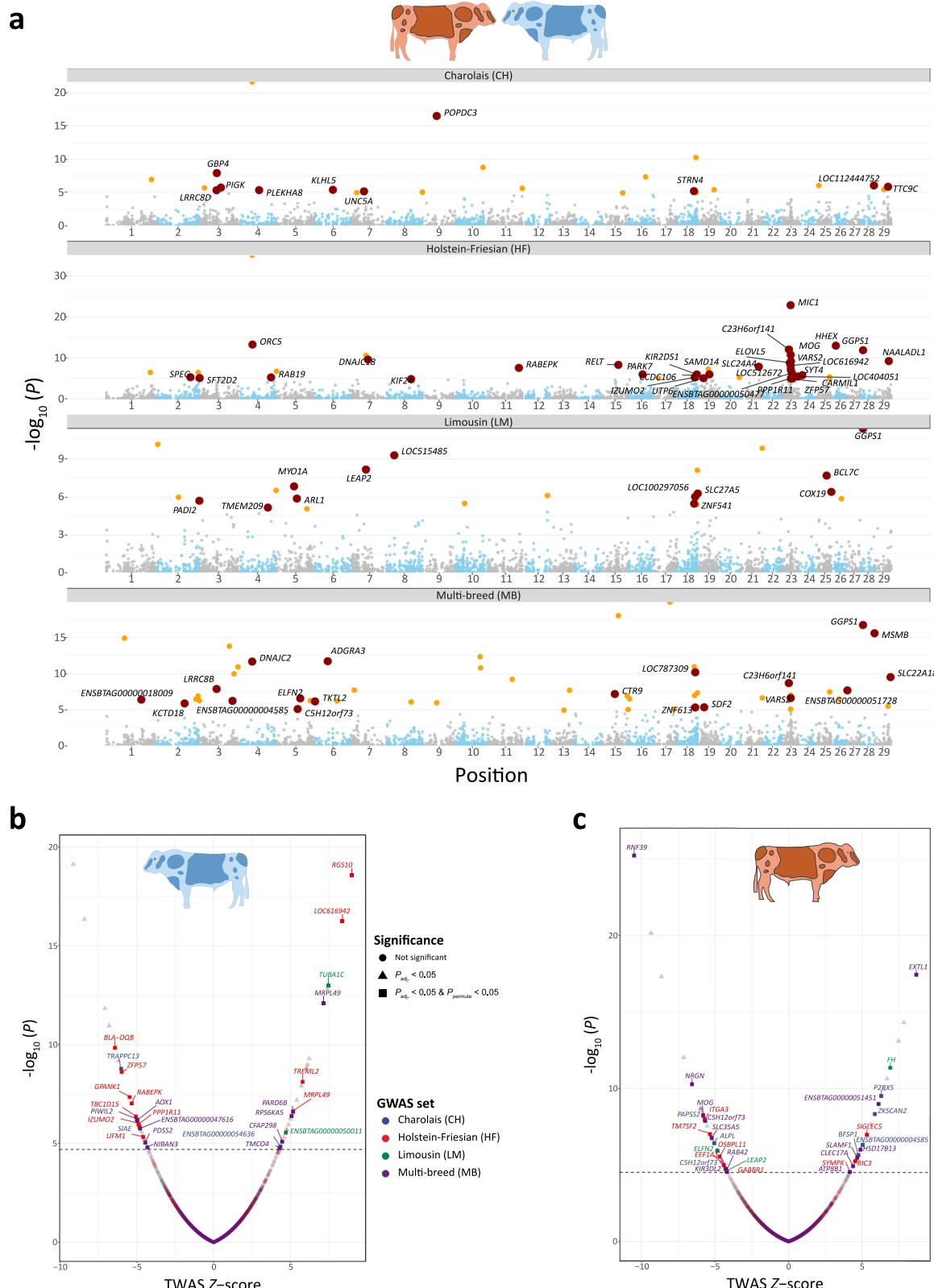

genes/pathways are substantially perturbed during mycobacterial infection and TB disease[87]. Previous work has shown that the cell subpopulations between bTB+ and bTB− cattle are significantly different[61] and this may have contributed to the transcriptional differences between the two groups. Once appropriate PB single-cell RNA-seq data from cattle infected with *M. bovis* becomes available, computational deconvolution[88] of this bulk RNA-seq data would enable the characterisation of cell type differences

between these two groups and attribute the gene expression changes to an increase or decrease in the abundance of a particular cell type. In this regard, the Cattle Cell Atlas[89] will be a significant resource for this endeavour.

Peripheral blood DEGs have recently been characterised as reflecting disease-induced expression perturbations rather than mechanistic disease-causing changes[90] and understanding the genetic architecture regulating transcriptional perturbations can shed light on the immunogenetics of bTB

**Fig. 5 | Transcriptome-wide association analysis (TWAS) results. a** Manhattan plot showing all TWAS associations for expression models generated in the analysis of all animals combined and imputed into four GWAS data sets (Charolais (CH), Holstein-Friesian (HF), Limousin (LM), and multi-breed (MB)). Yellow data points have a Bonferroni FDR $P_{adj.} < 0.05$, and red points correspond to genes that have a Bonferroni $P_{adj.} < 0.05$, and $P_{perm.} < 0.05$. Labelled genes correspond to red data points in the plot. **b** Volcano plot highlighting significant TWAS associations for expression models generated in the control group (bTB−). The x-axis indicates the TWAS Z-score, and the y-axis shows the nominal ($-\log_{10}$ scale) P-value of association. Associations are coloured based on the GWAS data set for which the expression model was imputed. Associations are shaped according to whether they

had a Bonferroni $P_{adj.} > 0.05$ (circle), $P_{adj.} < 0.05$ (triangle), or $P_{adj.} \leq 0.05$ and $P_{perm.} < 0.05$ (square). The dashed line corresponds to a Bonferroni $P_{adj.}$ cut-off ($P < 2.00 \times 10^{-5}$). **c** Volcano plot highlighting significant TWAS associations for expression models generated in the reactor group (bTB+). The x-axis indicates the TWAS Z-score, and the y-axis shows the nominal P-value of association. Associations are coloured based on the GWAS data set for which the expression model was imputed. Associations are shaped according to whether they had an FDR $P_{adj.} > 0.05$ (circle), $P_{adj.} < 0.05$ (triangle), or $P_{adj.} \leq 0.05$ and $P_{perm.} < 0.05$ (square). The dashed line corresponds to a Bonferroni $P_{adj.}$ cut-off ($P < 3.14 \times 10^{-5}$). The figure legend for (**b**, **c**) is common to both.

disease. To this end, our cis-eQTL analysis highlighted hundreds of thousands of cis-eVariants that were associated with thousands of cis-eGenes (Table 1). Similarly, and as expected, we also noted that our ability to detect cis-eQTLs with small effects ($|slope\_aFC| < 1$) was also dependent on sample size (Fig. 4a). We also showed that there are multiple independent cis-eQTLs acting on thousands of genes (Table 1 and Fig. 3b) indicating that there is substantial allelic heterogeneity present in regulating the expression of genes in non-infected control and *M. bovis*-infected cattle. Although PB is cellularly heterogeneous, we obtained good replication of cis-eQTLs in an external cohort from the Cattle GTEx Consortium[54] using AC, Storey's $\pi_1$ statistic, and Spearman correlation of effect size estimates (Fig. 3e, f). We noted higher replication rates (Storey's $\pi_1$ statistic) in the bTB+ cohort hypothesised to be due to the high proportion of large-effect cis-eQTLs identified in this cohort given the small sample size ($n = 60$), with this observation being documented previously by the human GTEx Consortium[91].

Our ieQTL analysis revealed a total of five cis-eQTLs displaying an interaction effect (FDR $P_{adj.} < 0.25$) with bTB status. Detecting higher-order trait-associated ieQTLs is challenging as Kasela et al.[77] reported that at an FDR $P_{adj.} < 0.25$, a total of 277 ieQTLs were identified across three major phenotypes (age, smoking and sex) using a sample size of circa. 900 individuals. Similarly, at an FDR < 0.05 in 5254 individuals, Yao et al.[76] characterised only 14 and 10 cis-eQTLs interacting with age and sex, respectively. Nonetheless, the top ieQTL identified here was the SNP Chr5:25861761:G:A associated with the poly(rC)-binding protein 2 gene (*PCBP2*; $P_{adj.} = 7.8 \times 10^{-5}$). *PCBP2* binds to poly(C) stretches of DNA and RNA and negatively regulates cyclic GMP-AMP synthase (cGAS) activity[92]. Cyclic-GAS is a cytosolic DNA sensor that, upon binding to DNA, converts ATP and GTP to cyclic GMP-AMP (cGAMP), which subsequently binds to and activates STING thereby triggering type I interferon production[93,94]. For tuberculous mycobacteria, cGAS and subsequent intracellular innate immune response mechanisms are activated by the mycobacterial ESX-1 secretion system[95]. Overexpression of *PCBP2* has been shown to reduce cGAS-STING signalling with the opposite pattern observed for loss of *PCBP2*[92]. Our results indicate that animals carrying a copy of the alternative (A) allele have lower expression of *PCBP2*, an effect that is flipped in the bTB− cohort (Fig. 4c). Taking these findings into account, bTB+ animals heterozygous or homozygous for the alternative allele may have a more robust innate immune response during *M. bovis* infection compared to GG homozygotes.

While nearly all expressed genes appear to have a cis-eQTL in a relevant context/tissue[91], we demonstrated that certain cis-eVariants have context-specific effects in bTB+ and bTB− cattle with key immune genes implicated such as *IFI16* and *IFITM3* (Fig. 4e, f). *IFI16* encodes a cytosolic DNA sensor and has been shown to be activated in murine bone marrow-derived macrophages (BMDMs) infected with *M. tuberculosis* and separately with *M. bovis*, with stunted IFN-β production observed in *IFI16*<sup>−/−</sup> BMDMs suggesting that *IFI16* plays a key role in host response to *M. tuberculosis* and *M. bovis* infection[96,97]. The observation that the variant Chr3:10984726:G:A is associated with a significant reduction in the expression of this gene in the bTB+ cohort (Fig. 4e) suggests that heterozygous or homozygous animals for the alternative allele may have a curtailed ability to produce IFN-β in response to *M. bovis* infection. Similarly, we observed that the variant

Chr29:50294904:G:A was associated with decreased expression of the IFN-induced transmembrane 3 gene (*IFITM3*) in the bTB+ cohort (Fig. 4f). The IFITM3 protein was previously shown to co-localise with early and late *M. tuberculosis* phagosomes contributing to endosomal acidification and destruction of the bacilli with knockdown of IFITM3 resulting in enhanced intracellular growth of *M. tuberculosis* in human monocytes[98]. Expression of *IFITM3* was also significantly downregulated in the bTB+ cohort (Supplementary Table S5), consistent with previous observations in humans using RT-qPCR[99]. Taken together, our results indicate that selecting against the major alternative allele (A) may have beneficial effects for the host in terms of increasing the expression of *IFITM3* enabling it to more effectively clear invading *M. bovis* bacilli during bTB disease through enhanced acidification of the phagosome. However, while the context-specific variants identified in this analysis do not equate to direct response-eVariants as the animals in the bTB− and bTB+ cohorts are different, future experiments with paired animals before and after *M. bovis* infection could enable validation of these context-specific cis-eVariants as true response-eVariants. Additionally, the cis-ieQTLs (Table 2) and context-specific cis-eVariants (Fig. 4d) identified in this study represent a candidate set of variants to test for a causal relationship with bTB susceptibility using integrative strategies such as colocalization[100] and summary-based Mendelian randomisation (SMR)[101] once a more comprehensive GWAS data set for bTB susceptibility, which includes allele frequency information becomes available.

Cis-eQTLs explain a small proportion of expression heritability whereas trans-eQTLs have been estimated to contribute up to 70% of the inter-individual variance in gene expression[58,102] and tag important genomic regulatory elements and transcriptional regulators (e.g. TFs/coTFs and miRNAs), which will be important for bridging the genome to phenome gap in livestock species[103]. Given the moderate sample sizes used for eQTL detection in this study, we did not map trans-eQTLs. However, our data will assist the Cattle GTEx Consortium in its efforts to map distal and inter-chromosomal trans-eQTLs in blood[54].

To identify genes with expression patterns correlated with bTB susceptibility, we integrated our molecular phenotype data and GWAS summary statistics for bTB susceptibility to conduct a TWAS. Our TWAS analyses revealed a total of 115 genes associated with susceptibility to bTB disease in cattle, many of which were breed- and expression model cohort-specific, an observation that reflects the polygenicity of this phenotypic trait[33] (Supplementary Fig. S13). However, several of the genes we identified have documented roles in the host response to MTBC infection and the immunopathology of TB disease. For example, in our AAG:CH TWAS cohort, we identified the guanylate binding protein 4 gene (*GBP4*; $P = 1.23 \times 10^{-8}$; $Z = −5.7$) as being significantly associated with bTB disease susceptibility (Fig. 5a). *GBP4* is an interferon-inducible gene that is upregulated and contributes to the Type 1 immune response during *M. tuberculosis* infection[104]. Additionally, expression of *GBP4* was shown to be significantly upregulated at +1 week, +2 weeks, and +10 weeks in blood samples from cattle experimentally infected with *M. bovis* and stimulated with PPD-b compared to control non-stimulated blood samples[71] and in PB leucocytes of *M. bovis*-infected cattle compared to non-infected control animals[65]. The most significant gene associated with bTB disease susceptibility in our bTB− group was the regulator of G-protein signalling 10 gene (*RGS10*; $P = 2.7 \times 10^{-19}$; $Z = 9.0$), which was identified in the

Holstein-Friesian GWAS panel (Fig. 5b). *RGS10* encodes an important anti-inflammatory protein and has previously been implicated in vitro in regulating macrophage activity, specifically limiting activation of the NF-κB pathway, reducing expression of tumour necrosis factor (TNF), and regulating macrophage M1–M2 repolarisation[105]. In murine models challenged with a lethal dose of influenza A virus, loss of *RGS10* resulted in increased cytokine and chemokine activity, and more pronounced recruitment of neutrophils and monocytes to the site of infection[106].

Members of the MTBC, including *M. tuberculosis* and *M. bovis*, have evolved a diverse range of strategies to modulate, subvert, and evade host innate immune responses and an important facet of this is the manipulation of granuloma formation and function[107]. Recent multi-modal profiling of the granuloma in cynomolgus macaques (*Macaca fascicularis*) experimentally infected with *M. tuberculosis* has shown that high-burden granulomas are characterised by Type 2 immunity and tissue-protective responses that maintain essential tissue functionality while paradoxically creating a niche for bacterial persistence, whereas low *M. tuberculosis* burden granulomas are governed by adaptive Type 1–Type 17 (Th1–Th17) and cytotoxic T cell responses that kill and destroy invading bacilli[108]. In this regard, we also identified the RELT TNF receptor gene (*RELT*; $P = 4.4 \times 10^{-9}$; $Z = 5.9$) in our AAG:HF TWAS cohort as being significantly associated with bTB disease susceptibility. *RELT* is a member of the TNF gene superfamily and evidence suggests that RELT may promote an immunosuppressive environment through suppression of IFN-γ, TNF, and IL-5 production in CD4[+] and CD8[+] T cells[109]. The triggering receptor expressed on myeloid-like cells 2 gene (*TREML2*) was also significantly associated with bTB susceptibility in our bTB−:HF TWAS cohort ($P = 7.6 \times 10^{-9}$; $Z = 5.8$). In monocytes infected with *M. tuberculosis*, overexpression of *TREML2* was shown to promote *IL6* transcription through activation of STAT3 and to suppress the Th1 T-cell response[110]. Expression of IL-6 induced by *M. tuberculosis* infection was also shown to inhibit the macrophage response to IFN-γ[111] and impaired intracellular killing of mycobacteria[112].

Taken together, based on our TWAS results, we can therefore hypothesise that the combined increased and decreased expression of several immunoregulatory genes dampens the proinflammatory immune response during *M. bovis* infection, suppresses the Th1 T-cell response and contributes to macrophage M2 polarisation and Type 2 immunity characteristics that lead to bTB disease susceptibility, bacterial persistence, pathogenesis, and disease. Our results are supported by a recent time series experiment of cattle experimentally infected with *M. bovis* whereby at 2 weeks post-infection, significant DEGs in comparison to −1 week pre-infection formed part of the glucocorticoid receptor signalling pathway that was hypothesised to upregulate and downregulate the expression of anti- and proinflammatory genes, respectively[69]. Overall, the TWAS genes identified here should be prioritised for further downstream functional analysis and the *cis*-eVariants associated with these genes may be incorporated as prior information in future genome-enabled breeding programmes for bTB disease susceptibility traits[113,114].

Although the TWAS approach is being increasingly applied to complex traits in plant and animal species, the statistical framework underpinning the methodology has come under criticism due to an inflated type 1 error rate as a consequence of failing to account for predictive expression model uncertainty in the gene-trait association test[115]. Stringent gene filtering through the application of a two-step statistical significance process with a Bonferroni $P_{adj.} < 0.05$ threshold followed by a post-hoc permutation test ($P < 0.05$) appears to control for this inflated false positive rate[115]. However, the permutation scheme itself is highly conservative and as such, true causal genes associated with the trait of interest may be filtered out owing to insufficient power[49]. Conversely, as animals in the bTB+ reference panel have a confirmed bTB diagnosis and are maintained for bTB diagnostics potency testing, it is possible that some of the results from the TWAS may be confounded by horizontal pleiotropy owing to the same causal variant having independent effects on both expression and the trait[49]. We would therefore prioritise significant TWAS associations identified in the analysis of the bTB− and AAG cohorts for further downstream analyses. Lastly, it is challenging to evaluate causality from our TWAS results due to issues associated with the sharing of GWAS variants between expression models, coregulation of a putatively causal and non-causal gene(s), and correlation of predicted expression models between tested genes[50]. A combination of a larger tissue/cell-specific reference panel in conjunction with other techniques such as colocalization[100] and Mendelian randomisation[116] would facilitate this approach.

## Methods

### Ethics statement

We have complied with all relevant ethical regulations for animal use. In this regard, all experimental procedures involving animals were conducted under ethical approval from the University College Dublin (UCD) Animal Research Ethics Committee (AREC-19-09-MacHugh) and experimental license AE18982/P141 from the Irish Health Products Regulatory Authority (HPRA) in accordance with the Cruelty to Animals Act 1876 and in agreement with the European Union (Protection of Animals Used for Scientific Purposes) regulations 2012 (S.I. No. 543 of 2012).

### Animal recruitment, sampling, and data acquisition

A total of $n = 60$ cattle infected with *M. bovis* and $n = 63$ non-infected control animals were recruited for the purpose of this study. All animals were male and were born between 2014 and 2019. The *M. bovis*-infected cattle (bTB+) were selected from a panel of naturally infected animals maintained for ongoing tuberculosis surveillance at the Department of Agriculture, Food, and the Marine (DAFM) Backweston Laboratory Campus farm (Celbridge, Co. Kildare, Ireland). These animals were skin tested by an experienced veterinary practitioner and had positive single intradermal comparative tuberculin test (SICTT) results where the skin-fold thickness response to purified protein derivative (PPD)-bovine (PPDb) exceeded that of PPD-avian (PPDa) by at least 12 mm. As an ancillary diagnostic test carried out in series, all animals were tested for *M. bovis* infection using the whole blood IFN-γ release assay (IGRA) test (Bovi-GAM®—Prionics AG, Switzerland) following the procedure described by Clegg et al.[72]. The criterion for IFN-γ test positivity was a test result difference greater than 80 ELISA units for the purified protein derivative (PPD)-bovine (PPDb) IFN-γ value minus the PPD-avian (PPDa) IFN-γ value (ΔPPD)[72]. Skin testing and blood sampling were conducted under statutory regulations according to the European Union (EU) Council trading Directive 64/432/EEC. Following the study, the bTB+ animals were removed by DAFM to a licensed commercial abattoir where they were slaughtered humanely as per national and EU regulations. Carcasses were examined for TB-like lesions by DAFM Veterinary Inspectors and recorded on national databases. During post-mortem examination, all the animals disclosed multiple lesions consistent with bovine tuberculosis. The non-infected control animals (bTB−) were selected from bTB-free cattle herds (all SICTT negative) with no recent history of *M. bovis* infection and were not sacrificed as part of this study. For the purposes of this study, the experimental designation of an animal as bTB− or bTB+ was based on their SICTT result. This is because the SICTT is more specific than the IGRA test and the positive predictive value (i.e. the probability that an animal testing positive from the SICTT is truly infected) is estimated to be 91.8%[117].

Peripheral blood was sampled from each animal using blood collection tubes with blood harvested from the tail vein. All tubes were inverted 4–6 times immediately after sampling and transported to the laboratory in a refrigerated cool box within 3 h of collection. Whole blood intended for isolation of total RNA was collected in Tempus™ blood RNA stabilisation tubes (Thermo Fisher Scientific) and these were stored at −80 °C prior to isolation and purification. A single heparin-coated tube was also collected from each animal (bTB+ and bTB−) for same-day IGRA testing to confirm infection status at the UCD Tuberculosis Diagnostics and Immunology Research Centre. Further details on genomic and transcriptomic data acquisition are detailed in Supplementary Note 1.

## Basic population genomics analysis and imputation

Genomic DNA (Supplementary Note 1) from all animals were genotyped using the Affymetrix Axiom™ Genome-Wide BOS-1 Array (Thermo Fisher Scientific) with SNP positions originally mapped to the UMD3.1 bovine reference genome assembly[118]. The CEL files were imported into Axiom Analysis Suite software tool v.5.1.1.1 following the Axiom Best Practices Genotyping Analysis Workflow with the required sample attributes[119]. The SNPolisher Recommended Probesets were exported in PLINK format annotated with the Axiom_GW_Bos_SNP_1.na35.annot.db annotation database using genome version UMD3.1 and NCBI version 6. This analysis yielded a total of 591,947 SNPs (91.23%) for downstream analyses. Prior to the remapping of SNPs to the ARS-UCD1.2 genome build[120], the genetic structure and diversity of the study population were evaluated as follows. PLINK v1.90b6.25[121] was used to filter out SNPs with a minor allele frequency (MAF) < 10%; that deviated from Hardy–Weinberg equilibrium (HWE; $P < 1 \times 10^{-6}$); and with a call rate < 0.95. The PLINK *--indep-pairwise* command was then used to prune variants in linkage disequilibrium (LD) with the following parameters: window size = 1000 kb; step size = 5 variants; and $r^2 > 0.2$. Following these steps, there were 34,272 SNPs available for examination of genetic structure using ADMIXTURE v.1.3[122] and principal component analysis (PCA) using PLINK.

The ADMIXTURE analysis was performed with the *--cv* option such that setting the number of ancestral populations to $K = 2$ produced the lowest cross-validation error. We then used pophelper v.2.3.1[123] to generate a structure plot. For the PCA analysis, we used the *--pca* function in PLINK and used a custom R v.4.3.2[124] script to plot the PCA results using ggplot2 v.3.4.1[125].

For the imputation up to whole-genome sequence (WGS) scale data, raw genotyped variants were first remapped from UMD3.1 to the ARS-UCD1.2 bovine genome assembly (Supplementary Note 2). Following this, a global cattle reference panel from Dutta et al.[73] was used as the imputation reference panel, which comprised a total of 10,282,187 SNPs derived from $n = 287$ distinct animals spanning a diverse range of breeds and geographic locations (55 populations: 13 European, 12 African, 28 Asian, and two Middle Eastern). Imputation was performed using Minimac4 v.1.03[126] with default parameters to impute the target genotype data set up to WGS, which resulted in a master imputed data set consisting of all $n = 123$ animals with genotypes for 10,282,037 SNPs (Supplementary Note 2).

## Transcriptomics data quality control, read alignment, and read mapping

The paired-end RNA-seq FASTQ files ($n = 123$; 60 bTB+ and 63 bTB−) were assessed using FastQC v.0.11.5[127], which showed that the RNA-seq data set was of sufficiently high quality to negate the requirement for hard or soft trimming. Following this, RNA-seq reads were aligned to the ARS-UCD1.2 bovine reference genome using STAR v.2.7.1a[128]. Read counts for each gene were then quantified using featureCounts v.2.0.6[129] and the ARS-UCD1.2 ensemble annotation file (https://ftp.ensembl.org/pub/release-110/gtf/bos_taurus/Bos_taurus.ARS-UCD1.2.110.gtf.gz) excluding chimeric fragments, aligning reads in a reversely stranded manner, and considering only fragments with both ends successfully aligned for quantification.

## Missing data imputation and sample mismatch assessment

Control sample C028 did not have any date of birth information available (Supplementary Table S1). Therefore, we inferred the age of C028 as the mean of all other animals that were sampled on the same date (02/05/2017). Control samples C039 and C041 were assigned the same animal identification number; therefore, to ensure that these animals were not duplicates, we estimated the identity-by-state (IBS) distance values between all samples by using the pruned SNPs prior to imputation to identify and remove duplicate animals using PLINK. The IBS distance values were calculated as:

$$IBS\ distance = (IBS2 + 0.5 \times IBS1)/(IBS0 + IBS1 + IBS2)$$

where *IBS0* is the number of IBS 0 non-missing variants, *IBS1* is the number of IBS 1 non-missing variants and *IBS2* is the number of IBS 2 non-missing variants. Sample pairs with IBS distance values > 0.85 were considered duplicates and only one sample was retained for subsequent analyses[54].

To ensure that the transcriptomics data and genome-wide SNP data for all 123 animals (bTB− and bTB+) were matched, we assessed the genotype consistency using the match BAM to VCF (MBV) function[130] that is part of the QTLtools (v 1.3.1) software package[131]. Briefly, MBV reports the proportion of heterozygous and homozygous genotypes (for each sample in a VCF file) for which both alleles are captured by the sequencing reads in all BAM files. Correct sample matches can then be verified, as they should have a high proportion of concordant heterozygous and homozygous sites between the genotype data and the mapped sequencing reads.

## Differential expression analysis

A differential expression analysis (DEA) was conducted between the reactor (bTB+) and control (bTB−) animal groups using DESeq2 v.1.40.2[132] and a design matrix, which included the following covariates: age in months, RNA-seq sequencing batch, and genetic structure in the form of PC1 and PC2 from the PCA of the pruned SNP data set prior to imputation with reactor status as the variable of interest. The PC1 and PC2 covariates were included because the crossbred/multibreed nature of the animals in our study population should be incorporated in the DEA contrast for the bTB+ and bTB− groups. Genes with raw expression counts ≥ 6 in at least 20% of samples were retained prior to the DEA. For the DEA, the null hypothesis was that the logarithmic fold-change (LFC) between the bTB+ and bTB− groups, for the expression of a particular gene is exactly 0. To account for the potential heteroscedasticity of LFCs, we implemented the approximate posterior estimation for generalised linear model coefficients (APEGLM) method[133] using the *lfcShrink* function. Genes with a Benjamini-Hochberg (BH) false discovery rate (FDR) adjusted $P$-value[134] ($P_{adj.}$) < 0.05 and a LFC > 0 or < 0 were considered significantly differently expressed genes (DEGs).

## *Cis*-eQTL mapping

For the mapping of *cis*-eQTLs, we used the human GTEx Consortium[135] pipeline with some minor modifications. We mapped *cis*-eQTLs in each group (bTB− and bTB+) separately to identify context-specific *cis*-eVariants, and we mapped *cis*-eQTLs in the merged AAG group to maximise our power and to characterise interaction *cis*-eQTLs (ieQTLs) associated with bTB status. Raw RNA-seq read counts were normalised using the trimmed mean of the M values (TMM) method[136] and the expression values for each gene were then inverse normally transformed across samples to ensure the molecular phenotypes followed a normal distribution. Genes with a raw expression count ≥ 6 and a transcript per million (TPM)[137,138] normalised expression count ≥ 0.1 in at least 20% of samples were retained for the eQTL analysis. For each group, we used the PCAForQTL R package v.0.1.0[139] to identify hidden confounders in the normalised and filtered expression matrices as it performed more effectively than the PEER R package[140] in comparative benchmarking[139]. The number of latent variables selected was determined using the elbow method via the *runElbow* function in PCAForQTL. We then merged these inferred covariates with known covariates (the top five genotype PCs of the imputed data set, age in months, sequencing batch, and infection status, where applicable) and removed highly correlated known covariates captured well by the inferred covariates (unadjusted $R^2 \geq 0.9$) using the PCAForQTL *filterKnownCovariates* function.

For the *cis*-eQTL mapping procedure, we used TensorQTL v.1.0.8[141]. We defined the *cis* window as ± 1 Mb from the transcriptional start site (TSS) of a gene. To identify significant *cis*-eQTLs, we invoked the permutation strategy in TensorQTL[142] to estimate variant-phenotype-associated empirical $P$-values with the parameter --mode *cis* to account for multiple variants being tested per molecular phenotype. We then used the Storey and Tibshirani FDR procedure[143] to correct the *beta* distribution-extrapolated empirical $P$-values to account for multiple phenotypes being tested genome-wide. A gene

with at least one significantly associated *cis*-eQTL was considered a *cis*-eGene.

To identify significant *cis*-eVariants associated with detected *cis*-eGenes, we followed the procedure implemented by the Pig GTEx Consortium[144]. Briefly, we first obtained nominal *P*-values of association for each variant-gene pair using the parameter --mode *cis_nominal*. We then defined the empirical *P*-value of a gene which was closest to an FDR of 0.05 as the genome-wide empirical *P*-value threshold (*pt*). Next, we calculated the gene-level threshold for each gene from the beta distribution by using the *qbeta(pt, beta_shape1, beta_shape2)* command in R with beta_shape1 and beta_shape2 being derived from TensorQTL. Variants with a nominal *P*-value of association below the gene-level threshold were included in the final list of variant-gene pairs and were considered significant *cis*-eVariants.

Following the Pig GTEx Consortium[144], to identify genes with multiple independent-acting *cis*-eQTLs, we performed a conditional stepwise regression analysis using the parameter --mode *cis_independent*. Briefly, the most significant variant was considered a putative *cis*-eQTL if it had a nominal *P*-value below the genome-wide FDR threshold inferred above. Next, using a forward stepwise regression procedure, the genotypes of this variant were residualized out from the phenotype quantifications and the process of regression, selection, and residualization was repeated until no more variants were below the *P*-value threshold resulting in *n* independent signals per gene. Finally, using backward stepwise regression, nearby significant variants were assigned to inferred independent signals.

To estimate the biological effect sizes of significant *cis*-eQTLs, we computed the log allelic fold-change (slope_aFC) with aFC.py (https://github.com/secaste/aFC)[74] for the top variant of each *cis*-eGene in each group. To plot the results of the *cis*-eQTL analyses, we extracted the residualized expression levels of genes using the QTLtools (v. 1.3.1) *correct* function[131] specifying the phenotype .bed file with the parameter --*bed* and the covariates .txt file with the --*cov* parameter. For genes of interest, we plotted the residualized expression values for each individual animal separated by their corresponding genotype as a boxplot.

## Interaction *cis*-eQTL mapping
To identify variants associated with bTB status, we performed an interaction analysis in the all animals group (AAG) cohort using TensorQTL accounting for the same covariates used in the initial *cis*-eQTL mapping procedure and fitting an interaction term between genotype and bTB status (g×i) as a binary variable. To identify genes with at least one significant interaction *cis*-eQTL (ieQTL), the top nominal ieQTL *P*-value for each gene was selected and corrected for multiple testing at the gene level using eigenMT[145]. Genome-wide significance was then determined by computing the BH-FDR on the eigenMT-corrected *P*-values. An ieQTL with an FDR $P_{adj.} < 0.25$ was considered to significantly interact with bTB status.

## Replication of *cis*-eQTLs
To assess the replicability of *cis*-eQTLs identified for each group in an independent cohort, we first downloaded blood *cis*-eQTL summary statistics (both permuted and nominal associations) from the Cattle GTEx Consortium (https://cgtex.roslin.ed.ac.uk/wp-content/plugins/cgtex/static/rawdata/Full_summary_statisitcs_cis_eQTLs_FarmGTEx_cattle_V0.tar.gz). We used three different measurements of agreement of eQTL effects when comparing eQTLs across the two studies: allele concordance (AC), Story's $\pi_1$ statistic, and Spearman correlation ($\rho$). AC provides an indication of the proportion of effects that have a consistent direction of effect (slope) within the set of eQTLs that is significant in both the discovery (the control bTB−, reactor bTB+, and combined AGG cohorts) and the replication cohort (the Cattle GTEx) and is expected to be 50% for random eQTL effects[146]. The parameter $\pi_1$[143] represents the proportion of true positive eQTL *P*-values in the replication cohort and is calculated as $1 - \pi_0$ (the proportion of true null eQTL *P*-values). The Spearman $\rho$ statistic estimates the correlation between the effect sizes (slope) of significant eQTLs in the discovery cohort and matched associations in the replication cohort, regardless of significance in the latter.

To calculate AC, we matched significant eQTLs in the discovery cohort to significant eQTLs in the replication cohort. We then calculated the proportion of these eQTLs that showed the same direction of effect. To calculate $\pi_1$, we obtained the *P*-values in the replication cohort of significant associations identified in the discovery cohort and used the *qvalue* function in R to estimate $\pi_0$. We then calculated $\pi_1$ as $1 - \pi_0$. Uncertainty estimates of $\pi_1$ were obtained using 100 bootstraps where SNPs were sampled with replacement and $\pi_1$ was recomputed each time[147]. To obtain the Spearman $\rho$ statistics, we calculated the Spearman correlation between significant eQTLs identified in this study to matched variant:gene pairs in the replication cohort, regardless of significance.

## Estimating the *cis*-SNP-based heritability of gene expression
To quantify the contribution of additive and dominance effects to the *cis*-SNP-based heritability of gene expression, for each normalised gene tested in the *cis*-eQTL analysis of the AAG cohort, we extracted the SNPs ± 1 Mb away from the TSS of the associated gene and constructed an additive genomic relationship matrix (GRM) using these SNPs via the --make-grm function in GCTA (v. 1.94.0)[148]. We then estimated the narrow-sense *cis*-heritability ($cis$-$h^2_{SNP}$) for each gene using a linear mixed model (LMM) accounting for the same covariates used in the *cis*-eQTL mapping procedure, with the variance components of the model being estimated using restricted maximum likelihood (REML) via the --greml function in GCTA[149]. We then conducted a likelihood ratio test (LRT) comparing the model with the additive GRM vs the null model of no genetic component and corrected the resulting *P*-values using the BH-FDR procedure. We defined genes with a $P_{adj.}$ value < 0.05 as being significantly heritable in this analysis. Following this, using the same *cis*-SNPs for each significantly heritable gene, we then calculated the dominance GRM using the --make-grm-d function in GCTA[150]. We then approximated the broad-sense cis-heritability ($H^2_{SNP}$) for each gene (i.e. the proportion of phenotypic variance explained by both dominant ($\delta^2_{SNP}$) and additive ($h^2_{SNP}$) genetic effects) by fitting an additive and dominant (AD) model using REML. We then used the LRT to evaluate whether the addition of the dominance component significantly improved the proportion of variation explained in the expression level of the gene and corrected resulting *P*-values using the BH-FDR procedure. For genes with a $P_{adj.}$ value < 0.05, dominance variance was considered to account for a significant proportion of the *cis* $H^2_{SNP}$ in expression levels.

## Testing the contribution of non-additive effects in *cis*-eQTL mapping
To examine what proportion of independent *cis*-eQTLs could be better explained by including a non-additive component in the linear model, we fitted two linear models for all independent *cis*-eQTLs across all three groups using the *lm()* function in R. The first model represented the same linear model used in the standard *cis*-eQTL mapping analysis with genotypes coded as 0, 1 and 2 depending on the number of alternative allele copies (nested). The second model was the same as the first model but also included a non-additive (dominance) component with both homozygous groups coded as 0 and heterozygotes coded as 1 (complex). For the independent *cis*-eQTLs that we could fit a dominance component for, we then used the *lrtest* function from the lmtest R package v. 0.9.40 to conduct an LRT between the complex and nested model to determine if the addition of the non-additive component in the linear model significantly improved model fit. The LRT statistic was compared to a chi-squared distribution with one degree of freedom and resulting *P*-values were corrected for multiple testing using the BH-FDR procedure. Complex models encompassing a non-additive component that achieved a $P_{adj.}$ value < 0.05 provided significant evidence for a deviation from additivity.

## GWAS data pre-processing
GWAS summary statistics for the present study were obtained from a single- and multi-breed GWAS experiment that leveraged WGS data from Run 6 of the 1000 Bull Genomes Project[151] as an imputation reference panel. The GWAS used estimated breeding values (EBVs) derived from an

*M. bovis* infection phenotype as the trait of interest for *n* = 2039 Charolais, *n* = 1964 Limousin, and *n* = 1502 Holstein-Friesian cattle[33]. Variants were remapped from UMD3.1 to ARS-UCD1.2 using a custom R script that was developed for a previous study that integrated the GWAS summary statistics with functional genomics data obtained from *M. bovis*-infected bovine alveolar macrophages (bAMs)[84]. To check for instances of strand flips, the reference and alternative allele pairs derived from Run 6 of the 1000 Bull Genomes Project were compared to the reference and alternative allele pairs in the ARS-UCD1.2 reference genome (https://sites.ualberta.ca/~stothard/1000_bull_genomes/ARS1.2PlusY_BQSR.vcf.gz). If a strand flip occurred, the beta values for each SNP were also inverted. A Wald-statistic *Z* score for each GWAS SNP was calculated by dividing the effect size (*β*) of a SNP by the standard error of the effect size.

### Transcriptome-wide association study (TWAS) analysis
Imputed genotype data for the three groups (bTB−, bTB+, and AAG) were converted to binary (.bed) format using PLINK with the *--keep-allele-order* parameter. The resulting files were then loaded into R using the bigsnpr v.1.10.8 and bigstatsr v.1.5.6 R packages[152]. Predictive models of expression for each gene were generated using the Mediator-enriched TWAS (MeTWAS) function within the MOSTWAS package v.0.1.0[59]. Briefly, MeTWAS first identifies an association in expression levels between a mediating biomarker (e.g. a transcription factor (TF)) and a gene of interest. It then builds a predictive model of expression for the mediating biomarker considering SNPs local to the biomarker. The predicted expression pattern of the biomarker (determined via five-fold cross-validation) is then included as a fixed effect with the effect sizes of putative mediators on the expression levels of the gene of interest estimated by ordinary least squares regression. Lastly, for the final predictive model of the gene of interest, the *cis*-eVariants are fitted as random effects using either elastic net regression or linear mixed modelling, whichever produces the highest five-fold McNemar's cross-validated adjusted $R^2$ value.

The mediating biomarkers used in MeTWAS included expressed regulatory proteins (TFs and co-TFs) curated from the AnimalTFDB database[153]. We first computed associations between mediating biomarkers and genes through correlation analysis with significant associations (BH-FDR < 0.01) being retained. We then retained mediating biomarker:gene associations in instances where the mediating biomarker was considered a *cis*-eGene. Genes that had significant non-zero heritabilities (nominal *P* < 0.05) for their expression levels, as computed by the LRT in GCTA[148] and for which MOSTWAS-derived predictive models achieved a five-fold McNemar's cross-validated adjusted $R^2$ value ≥ 0.01 were retained for the gene–trait association test. The maximum number of mediating biomarkers to include in the expression model for a gene was set to ten.

Within the MOSTWAS framework, expression models were imputed into the GWAS summary statistics using the ImpG-Summary algorithm[49,154] and a weighted burden *Z*-test was employed in the gene–trait association test[49,154]. Genes with a Bonferroni-adjusted *P*-value < 0.05 were considered candidate genes associated with bTB susceptibility. To assess whether the same distribution of GWAS SNP effect sizes could yield a significant association by chance, we implemented a permutation scheme on significant (Bonferroni-adjusted *P*-value < 0.05) TWAS genes where we sampled, without replacement, the SNP effect sizes 1000 times and recomputed the weighted burden test statistic to generate a permuted null distribution[49]. Genes with a permuted *P*-value < 0.05 were considered significantly associated with bTB disease status.

### Gene set overrepresentation and functional enrichment analyses
Gene set overrepresentation and functional enrichment analyses were conducted using a combination of the g:GOSt tool within g:Profiler v.0.2.2[155] and Ingenuity® Pathway Analysis—IPA® (Summer 2023 release; Qiagen). For IPA®, the target species selected included *Homo sapiens*, *Mus musculus*, and *Rattus rattus* with all cell types selected in addition to the Experimentally Observed and High Predicted confidence settings. We followed best practice recommendations to account for tissue-specific sampling biases in

gene set overrepresentation and functional enrichment analyses[156]. Consequently, for analysis of DEGs, the background set consisted of all expressed genes that were tested for differential expression. Additionally, to minimise the computational burden associated with IPA®, as input for the overrepresentation analysis of DEGs, we selected a subset of DEGs with an FDR $P_{adj.}$ value < 0.01. For analyses of our query gene sets, we selected the gene ontology biological process (GO:BP) and the cellular component (GO:CC)[157] databases in addition to the Kyoto Encyclopaedia of Genes and Genomes (KEGG)[158] and Reactome[159] repositories. To identify significantly enriched/overrepresented pathways, a BH-FDR multiple testing correction was applied (FDR $P_{adj.}$ < 0.05).

### Statistics and reproducibility
Statistical analyses and visualisations were conducted using R (v. 4.3.2). The statistical methods and tests employed for each analysis are explained in the text and figure legends, where applicable. A comprehensive description of the statistical methodology applied for each of the differential expressions, *cis*-eQTL mapping, and TWAS analyses, respectively, is available in the "Methods" section. All data-intensive computational procedures were performed on a 36-core/72-thread compute server (2× Intel® Xeon® CPU E5–2697 v4 processors, 2.30 GHz with 18 cores each), with 512 GB of RAM, 96 TB SAS storage (12 × 8 TB at 7200 rpm), 480 GB SSD storage, and with Ubuntu Linux OS (version 18.04 LTS).

### Reporting summary
Further information on research design is available in the Nature Portfolio Reporting Summary linked to this article.

### Data availability
The RNA-seq data from the 60 *M. bovis*-infected (bTB+) and 63 control (bTB−) cattle are available at the Gene Expression Omnibus (GEO) under the BioProject Accession GSE255724. The raw UMD3.1 high-density genotype data, ARS-UCD1.2 imputed and filtered data, *cis*-eQTL results for the three groups (bTB−, bTB+ and AAG), GWAS summary statistics for the four breed cohorts (CH, LM, HF, and MB) in addition to expression models generated for use in the TWAS are available at Zenodo[160] (https://doi.org/10.5281/zenodo.13752056). Of note, raw GWAS data were not analysed in this study; the original GWAS information can be obtained from the Irish Cattle Breeding Federation (ICBF) and additional information about sequence and genotype data availability is provided by Ring et al.[33]. Cattle GTEx Consortium *cis*-eQTL summary statistics can be obtained from https://cgtex.roslin.ed.ac.uk/wp-content/plugins/cgtex/static/rawdata/Full_summary_statisitcs_cis_eQTLs_FarmGTEx_cattle_V0.tar.gz.

### Code availability
The computer code and scripts used in this study are available at the following GitHub link: https://github.com/jfogrady1/Bovine_eQTL_TWAS and Zenodo[161].

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

## Acknowledgements

J.F.O.'G. was supported by Research Ireland through the Research Ireland Centre for Research Training in Genomics Data Science (grant no. 18/CRT/6214). This study was also supported by Science Foundation Ireland (SFI) Investigator Programme Awards to D.E.M. and S.V.G. (grant nos. SFI/08/ IN.1/B2038 and SFI/15/IA/3154); the University College Dublin—University of Edinburgh Strategic Partnership in One Health awarded to D.E.M., S.V.G., J.G.D.P., and E.L.C.; and the Department of Agriculture, Food and the Marine (DAFM) project awards to D.E.M. (TARGET-TB: grant no. 17/RD/US-ROI/52; TB-ORNOT-TB: grant no. 2023RP902). Note: Since the 1st of August 2024, Science Foundation Ireland (SFI) has been part of Taighde Éireann—Research Ireland (www.researchireland.ie). The authors would like to thank Mairead Doyle for their help in interpreting the interferon-γ release assay test results for the animals used in the study. The authors would also like to express their gratitude to Bojan Stojkovic for assistance with animal ethics and Di Wang for assistance with experimental work.

## Author contributions

J.F.O.'G., K.G.M., E.G., I.C.G., S.V.G., and D.E.M. conceived and designed the study. D.E.M., I.C.G., S.V.G., J.G.D.P., and E.L.C. acquired funding for the study. E.G., K.G.M. and M.M. facilitated access to animals for the study and C.N.C., G.P.M., S.L.F.O'D., J.A.W. and J.A.B. performed experimental work. J.F.O'G. performed all bioinformatics analyses and J.G.D.P., V.R., H.P., J.A.W., G.P.M., and T.J.H. provided important scientific input. J.G.D.P. and V.R. supplied the WGS Global Reference Panel and provided advice for variant remapping, strand flipping, and genomic imputation. H.P. contributed to the biological interpretation of results and discussions around *cis*-eQTL mapping. J.F.O'G. wrote the first draft of the manuscript, and prepared all figures, tables, and supplementary information with input from D.E.M. All authors read and approved the final manuscript.

## Competing interests

The authors declare no competing interests.
