## [Transparent Peer Review file · Communications Biology]

Integrative genomics sheds light on the immunogenetics of tuberculosis in cattle

Corresponding Author: Professor David MacHugh

This manuscript has been previously reviewed at another journal. This document only contains information relating to versions considered at Communications Biology.

Version 0:

Reviewer comments:

Reviewer #1

(Remarks to the Author)

O'Grady et al perform a multi-omic analysis of bovine tuberculosis (bTB) including genetic and transcriptomic data modalities. They identify differentially expressed genes and cis eQTLs associated with bTB as well as integrate these data to determine genes associated with bTB using a transcriptome-wide association study approach. Results are also then compared to public databases. Overall, the manuscript is well-written and clear. The analyses are thorough and founded on sound statistical principles. In particular, the authors' public code and detailed methods are to be commended. Some specific methods clarifications are needed, and there are a couple of opportunities for additional analyses that would further elevate this study. Please see specific comment below.

Major comments and questions

- Section starting line 330: Why was a subsample approach used instead of the interaction of genotype and bTB status? Is an interaction model what the author's refer to as a "standard response eQTL analysis" on line 347?
- Line 368-380: These associations should be further assessed for their potential causal relationship using mediation analysis of SNP->gene expression->bTB susceptibility.
- Figure 4: For SNP such as Figure 4E or G, was a dominant model tested to verify that low variance in a single homozygous genotype (because N is small) is not driving significant results?
- Line 743: Why were inferred covariates preferentially retained over known covariates? Wouldn't results be more interpretable using known covariates when available?
- Line 500: The authors outline a cell deconvolution analysis for which all the data are already available. Why was this analysis not performed and relevant cell proportions explored as covariates in models?
- Line 602-622: This paragraph begins with relevant discussion of the limitations of the TWAS approach. However, the authors then discuss several non-significant gene findings, hypothesizing that these genes did not meet significance criteria due to low statistical power. Just as possible, given a larger sample size, these genes would be confirmed as nonsignificant. There is no evidence in the current to support findings regarding these genes and interest in "immunologically relevant genes" is not at all specific to the few genes highlighted here. The authors "cherry picking" appears to be a priori interests that do not align with the genome-wide approaches used.

Minor comments

- Table 1: The distance column does not capture unique information per row. Thus, this information should be included in the Table title and this column removed.
- Figure S11: I believe loesse should be loess.
- Line 368-380: It is a bit difficult to follow the use of "upregulated" vs "downregulated" here. Suggest referring to DEGs as bTB- or bTB+ associated to clarify.
- Figure 5: Please verify that the cow icon color matches associated data. Based on the legend, I think that the control (blue) icon is on infected data.
- Line 507-511: This text is not necessary as it is stating a well-established statistical principle i.e. that sample size and effect size impact statistical power.

Reviewer #2

(Remarks to the Author)

This is an extensive study and generally well written.

Some comments:

Line 143-154: Move method description to methods section and retain only relevant results here.

Lines 150-154: We do know that many interactions occur at the biological level due to the influence of both internal and external factors. One of these factors could be responsible for the conflicting phenotypes by the different methods for these 3 individual animals so retaining them in the analyses could affect the analyses outcome. Instead, why not remove and try to understand the reason behind the conflicting results?

Figure 1: There was no mention of IFN- γ assay in the methods section.

Figure 1: SNP array data- Why did you not consider combining this with culled SNPs from the RNA-Seq data of the same animals? This strategy would make available more population specific SNPs and a better quality imputed variant set for downstream analysis. See Koyama et al., 2024. While most SNP arrays do present the opportunity to capture SNPs spaced at defined distances of the genome, they may not capture SNPs present in all genes nor population specific variants. Koyama, S., Liu, X., Koike, Y. et al. Population-specific putative causal variants shape quantitative traits. *Nat Genet* 56, 2027–2035 (2024). <https://doi.org/10.1038/s41588-024-01913-5>

Lines 224-227: It is not clear why only a subset of DEG was used for overrepresentation analysis.

Lines 363- 384: Results indicate that only few variants were found associated with the expression of some genes. I think these results could have been different if the SNPs generated by genotyping with a chip were those culled from the RNA-Seq data, or a combination of both strategies followed by imputation, thereby increasing the proportion of population specific variants. See line 553 and comment about this in the discussion section.

Line 445: Table 2?

Line 646: Were all the studied animals sacrificed in this study? If yes, provide further details.

Line 659: Consider revising as- Genomic DNA (supplementary note 1) from all animals were genotyped...

Line 816: Specify the three groups for clarity to readers and explain the rationale for the AAG group (were appropriate) since the aim of the study is finding factors explaining the most variance between the bTB- and bTB+ groups.

Discussion: Comment on the application of the findings vis-a-vis the objectives of the Cattle FarmGTEx project.

Version 1:

Reviewer comments:

Reviewer #1

(Remarks to the Author)

I appreciate the extensive analytical work undertaken by the authors to address my questions and concerns. I have no further requests.

Reviewer #2

(Remarks to the Author)

My comments have been well handled and I do not have further comments.

General response to reviewers.

AUTHOR: We would like to thank both Reviewers for their thoughtful and detailed evaluations of our manuscript and for providing valuable and insightful comments. We have carefully modified the manuscript accordingly. In addition to the point-by-point responses detailed below and as described in the marked-up document of the main manuscript file, in our review of the manuscript, we came across two typographical issues detailed below that we have now resolved. In addition, minor formatting and typographical edits to the text and figures are indicated in the marked-up revised Article and Supplementary Information files.

1. We incorrectly reported the number of conditionally independent, and consequently the total number of independent *cis*-eQTLs in **Table 1**. This was because we generated this table of results prior to setting the seed in our analysis to ensure reproducibility. The updated results of this Table and calculations derived from it reflect the results that are deposited in our Zenodo repository (<https://doi.org/10.5281/zenodo.13752056>).
2. We incorrectly reported the total number of bTB+ magnified, bTB- magnified, shared and non-significant *cis*-eVariants in Fig. 4c (**now corrected in Fig. 4d in the revised manuscript**) due to the same reason as above. The correct results were originally reported in **Supplementary Table S15**, and we have now updated this figure panel and reports in the manuscript to reflect this.

Reviewer #1 (Remarks to the Author):

O'Grady et al perform a multi-omic analysis of bovine tuberculosis (bTB) including genetic and transcriptomic data modalities. They identify differentially expressed genes and *cis* eQTLs associated with bTB as well as integrate these data to determine genes associated with bTB using a transcriptome-wide association study approach. Results are also then compared to public databases. Overall, the manuscript is well-written and clear. The analyses are thorough and founded on sound statistical principles. In particular, the authors' public code and detailed methods are to be commended. Some specific methods clarifications are needed, and there are a couple of opportunities for additional analyses that would further elevate this study. Please see specific comment below.

AUTHOR: The authors thank Reviewer 1 for taking the time to review our manuscript and for providing positive feedback and comments. We are glad that Reviewer 1 appreciates our public code and detailed methods and finds that the manuscript is well-written, clear and based on sound statistical principles.

Major comments and questions

Section starting line 330: Why was a subsample approach used instead of the interaction of genotype and bTB status? Is an interaction model what the author's refer to as a "standard response eQTL analysis" on line 347?

AUTHOR: The authors thank Reviewer 1 for this comment. We refer to a "standard response eQTL analysis" to that described by Umans, et al. ¹ whereby *cis*-eQTL mapping is performed both prior to and after the application of a stimulus/treatment in the *same* set of individuals. Loci associated with intrapopulation differences in expression before or after the stimulus/treatment, but not in both, or that display significant effects in the opposite direction are classified as response-QTLs (reQTLs). The study conducted by Barreiro, et al. ² illustrates the experimental design involved for what we termed a "standard response eQTL analysis" in the context of human tuberculosis (hTB) whereby eQTL

mapping was performed in $n = 65$ dendritic cells before and after *M. tuberculosis* infection in the same set of samples. As the animals from both cohorts (reactor (bTB+) and controls (bTB-)) in our analyses were not paired, we initially believed that this “standard response eQTL analysis” was not possible. Although we identified significant *cis*-eVariants in the control and reactor group but not both, we therefore made the decision to classify these as putative “context-specific *cis*-eVariants”.

We thank Reviewer 1 for mentioning the use of an interaction model, an aspect that we had not considered initially given the unpaired nature of the animals. While Umans, et al. ¹ describe that reQTLs display significant gene-environment interactions, we acknowledge that various interaction-based analyses have recently been performed to identify such effects in humans³, including experiments with unpaired data⁴. On reflection, based on this comment, we hypothesised that conducting an interaction analysis may be a useful strategy to identify interaction *cis*-eQTLs (ieQTLs) associated with bTB disease status.

We therefore performed an interaction analysis on the all animal group (AAG) cohort using TensorQTL accounting for the same covariates used in the initial *cis*-eQTL mapping procedure and fitting an interaction term between genotype and bTB status ($g \times i$) as the binary variable. To identify genes with at least one significant interaction *cis*-eQTL (ieQTL), the top nominal ieQTL *P*-value for each gene was selected and corrected for multiple testing at the gene level using eigenMT⁵. Genome-wide significance was then determined by computing Benjamini-Hochberg (BH) FDR on the eigenMT-corrected *P*-values. An ieQTL with an FDR $P_{adj.} < 0.25$ was considered significantly interacting with bTB status.

From this analysis, we identified two SNPs at the traditional adjusted 0.05 significance level (FDR $P_{adj.} < 0.05$) interacting with bTB status in the AAG cohort. However, we note that detecting higher-order trait associated ieQTLs is challenging as Kasela, et al. ³ reported that at an FDR $P_{adj.} < 0.25$, a total of 277 ieQTLs were identified across three major phenotypes (age, smoking and sex) using a sample size of ~900 individuals. Similarly, at an FDR < 0.05 in 5,254 individuals, Yao, et al. ⁶ characterised only 14 and 10 *cis*-eQTLs interacting with age and sex, respectively. Here, at an FDR $P_{adj.} < 0.25$, the number of SNPs significantly associated with bTB status increased to five (**Table R1, now Table 2 in revised manuscript**)

Table R1 (now Table 2 in revised manuscript): Significant (FDR $P_{adj.} < 0.25$) interaction *cis*-eQTL analysis results for the interaction term of genotype and bTB status ($g \times i$) fitted in the all-animal group (AAG) cohort using TensorQTL⁷. The *cis*-eVariant and corresponding *cis* gene are displayed, along with the nominal *P*-value of the interaction association, the gene-level eigenMT adjusted *P*-value⁵ and global false discovery rate (FDR) adjusted *P*-value.

Variant	Gene	P -value _{$g \times i$}	eigenMT $P_{adj.}$	FDR $P_{adj.}$
Chr5:25861761:G:A	PCBP2	4.9×10^{-12}	5.3×10^{-9}	7.8×10^{-5}
Chr18:49353483:G:C	PLD3	8.0×10^{-10}	8.8×10^{-7}	6.4×10^{-3}
Chr15:53330033:T:C	RELT	1.3×10^{-8}	1.7×10^{-5}	0.08
Chr3:53539162:G:T	GBP4	5.4×10^{-8}	6.8×10^{-5}	0.22
Chr23:10034540:C:T	PNPLA1	6.9×10^{-8}	7.6×10^{-5}	0.22

We have also modified Figure 4 to include a visualisation of the ieQTL for the gene *PCBP2* (Fig 4c in revised manuscript).

L426-445:

Modified version of Fig. 4: Effect of sample size on *cis*-QTL discovery and the characterisation of interaction *cis*-eQTLs and context-specific *cis*-eVariants. **a** Boxplot showing the distribution of the absolute \log_2 allelic fold change (slope_aFC) for top significant *cis*-eQTLs ($P_{\text{adj.}} < 0.05$) identified in the all animals group (AAG), the control group (bTB-) and the reactor group (bTB+) respectively. **b** Scatter plot showing the Spearman correlation (ρ) between the slope_aFC and the effect size from the linear regression model of top significant *cis*-eQTLs in the AAG, bTB-, and bTB+ cohorts respectively. **c** Visualisation of an interaction *cis*-eQTL (ieQTL) for the gene *PCBP2* identified in the AAG cohort. Data points are coloured and shaped based on an animal's experimental designation. Trend lines are inferred from setting the *geom_smooth* function in R setting the method to "lm". **d** Scatter plot illustrating the effect sizes of *cis*-eVariants tested in both the bTB- and bTB+ cohorts. *Cis*-eVariants are coloured depending on whether they are significant ($P_{\text{adj.}} < 0.05$) in the bTB- cohort only (blue), in the bTB+ cohort only (red), in both the bTB- and bTB+ cohorts (gold), or neither (grey). **e** Boxplot displaying the residualized expression values for the gene *IFL16*, with individuals separated based on their genotype at the SNP Chr3:10984726:G:A and coloured based on their respective cohort. Nominal *P*-values of association in each cohort are displayed with slope values from the linear regression model. **f** Same as **e** but for the gene *IFITM3* and the SNP Chr29:50294904:G:A. **g** Same as **e** but for the gene *IL1R1* and the SNP Chr11:6851248:C:G. **h** Same as **e** but for the gene *RGS10* and the SNP 26:40304855:C:A. The box plots in **a** and **e-h** cover the interquartile range with the median line denoted at the centre, and the whiskers extend to the most extreme data point that is no more than $1.5 \times \text{IQR}$ from the edge of the box.

We have also modified and expanded the Results section to reflect these new results and to clarify what we consider to be a "standard response-eQTL analysis" replacing the word "standard" with "traditional and paired":

L370 - L384: In this experiment, a traditional and paired response-eQTL analysis, which identifies eQTLs in the same set of subjects before or after the application of a stimulus/treatment but not in both, or that characterises eQTLs with opposing effects in both groups was not possible as different animals were present in the bTB- and bTB+ cohorts. However, to identify genetic variation that is associated with different expression levels in the bTB+ and bTB- groups, we conducted an interaction analysis and fitted an interaction term in the AAG cohort between the genotype at a locus and bTB status ($g \times i$) in the TensorQTL linear model accounting for the same covariates used in the initial *cis*-eQTL mapping procedure. From this analysis, we identified a total of five *cis*-eQTLs significantly (FDR $P_{adj.} < 0.25$) interacting with bTB status (**Table 2**). The genes associated with these ieQTLs included *PCBP2* (**Fig 4c**), *PLD3*, *RELT*, *GBP4* and *PNPLA1*. We note that previous studies have used this relaxed FDR $P_{adj.}$ cut-off, illustrating that identifying ieQTLs interacting with higher-order traits is challenging, but that failure to detect an ieQTL effect does not indicate absence of an ieQTL effect^{3,6}. Therefore, to identify putative context-specific *cis*-eVariants, we compared the effect of top *cis*-eQTLs for common genes and SNPs tested in both the bTB- and the bTB+ groups, because substantial differences in these effects may reflect or indicate context-specific regulatory mechanisms.

We have also included the following statement discussing our results in the Discussion section

L566 – L583: Our ieQTL analysis revealed a total of five *cis*-eQTLs displaying an interaction effect (FDR $P_{adj.} < 0.25$) with bTB status. Detecting higher-order trait associated ieQTLs is challenging as Kasela, et al. ³ reported that at an FDR $P_{adj.} < 0.25$, a total of 277 ieQTLs were identified across three major phenotypes (age, smoking and sex) using a sample size of circa. 900 individuals. Similarly, at an FDR < 0.05 in 5,254 individuals, Yao, et al. ⁶ characterised only 14 and 10 *cis*-eQTLs interacting with age and sex respectively. Nonetheless, the top ieQTL identified here was the SNP Chr5:25861761:G:A associated with poly(rC)-binding protein 2 gene (*PCBP2*; $P_{adj} = 7.8 \times 10^{-5}$). *PCBP2* binds to poly(C) stretches of DNA and RNA and negatively regulates cyclic GMP-AMP synthase (cGAS) activity⁸. Cyclic-GAS is a cytosolic DNA sensor that, upon binding to DNA, converts ATP and GTP to cyclic GMP-AMP (cGAMP), which subsequently binds to and activates STING thereby triggering type I interferon production^{9,10}. For tuberculous mycobacteria, cGAS and subsequent intracellular innate immune response mechanisms are activated by the mycobacterial ESX-1 secretion system¹¹. Overexpression of *PCBP2* has been shown to reduce cGAS-STING signalling with the opposite pattern observed for loss of *PCBP2*⁸. Our results indicate that animals carrying a copy of the alternative (A) allele have lower expression of *PCBP2*, an effect that is flipped in the bTB- cohort (**Fig. 4c**). Taking these findings into account, bTB+ animals heterozygous or homozygous for the alternative allele may have a more robust innate immune response during *M. bovis* infection compared to GG homozygotes.

We have also modified the Methods section to describe this analysis

L847 – L855:

Interaction *cis*-eQTL mapping

To identify variants associated with bTB status, we performed an interaction analysis in the all animal group (AAG) cohort using TensorQTL accounting for the same covariates used in the initial *cis*-eQTL mapping procedure and fitting an interaction term between genotype and bTB status ($g \times i$) as a binary variable. To identify genes with at least one significant interaction *cis*-eQTL (ieQTL), the top nominal ieQTL *P*-value for each gene was selected and corrected for multiple testing at the gene level using eigenMT⁵. Genome-wide significance was then determined by computing Benjamini-Hochberg (BH)

FDR on the eigenMT-corrected P -values. An ieQTL with an FDR $P_{\text{adj.}} < 0.25$ was considered significantly interacting with bTB status.

- Line 368-380: These associations should be further assessed for their potential causal relationship using mediation analysis of SNP->gene expression->bTB susceptibility.

AUTHOR: We thank Reviewer 1 for raising this point and for proposing a mediation analysis. We had considered conducting a summary-based mendelian randomisation (SMR) analysis¹² to determine if the effect sizes of *cis*-eQTLs for bTB susceptibility are mediated through gene expression. However, as we are dealing with GWAS summary statistics for bTB susceptibility, we do not have access to the underlying raw genotypic data and are unable to calculate the allele frequencies for the SNPs in the GWAS panels (these were not reported by Ring et al., (2019)¹³) that are required for this analysis. This is the primary reason why we focused on the TWAS integrative strategy. We had considered imputing the allele frequencies into the GWAS reference panels using the 1000 Bull Genomes Project as a reference. However, problems associated with out of sample allele frequency estimates, especially given the diverse geographical location and sparse breed information of the 1000 Bull Genomes samples, would not lead to reliable imputation of allele frequency estimates

Future work will involve obtaining access to raw genotype data from a larger set of sires phenotyped for the bTB susceptibility composite trait. This will enable the generation of new GWAS data that will facilitate the application of integrative genomics analyses including SMR and colocalization using COLOC¹⁴ (which also requires allele frequency information) to tease apart the causal mechanisms underpinning bTB susceptibility. In this regard, performing a summary-based MR analysis using our interaction *cis*-eQTLs (**Table R1, Table 2 in revised manuscript**) and context-specific *cis*-eVariants (**Fig. 3d**) would be the first application using this new GWAS data.

We have modified the Discussion section to highlight this

L606 – L611: Additionally, the *cis*-ieQTLs (**Table 2**) and context-specific *cis*-eVariants (**Fig. 4d**) identified in this study represent a candidate set of variants to test for a causal relationship with bTB susceptibility using integrative strategies such as colocalization¹⁴ and summary-based Mendelian randomisation (SMR)¹² once a more comprehensive GWAS data set for bTB susceptibility, which includes allele frequency information becomes available.

- Figure 4: For SNP such as Figure 4E or G, was a dominant model tested to verify that low variance in a single homozygous genotype (because N is small) is not driving significant results?

AUTHOR: We thank Reviewer 1 for raising this interesting issue, an aspect that we had not considered given that the majority of *cis*-eQTLs display additive effects¹⁵, with dominance effects predominantly confined to eQTLs acting in *trans* in mammals¹⁶. Additionally, for complex traits in humans, dominance variation at common SNPs contributes a small proportion to overall phenotypic variability, and analysing trait variability due to genomic variability at a locus has been shown to be robustly supported by an additive model^{17,18}. However, evidence for dominance variation contributing to the *cis* heritability of gene expression in cattle is sparse.

However, we believe that there is scope for a wider analysis of dominance effects to the *cis*-heritability of gene expression and *cis*-eQTL mapping. We therefore aimed to address this comment via two strategies that we think can enhance the scientific findings of the paper; 1) to determine whether there is evidence for dominance variation contributing to the *cis*-heritability of gene expression in the AAG

cohort and; 2) to determine if adding a non-additive component into the *cis*-eQTL linear model significantly enhanced the proportion of variation explained in gene expression versus using an additive component alone in all three groups (bTB-, bTB+ and AAG).

1. Testing contribution of dominance effects to *cis*-heritability of gene expression

We first considered whether there was evidence of non-additive genetic variance components (i.e. dominance) contributing to the heritability of gene expression in cattle using the AAG cohort ($n = 123$). To do this, for each normalised gene tested in the *cis*-eQTL analysis, we extracted the SNPs ± 1 Mb away from the transcriptional start site of the associated gene and constructed an additive genomic relationship matrix (GRM) using these SNPs via the `--make-grm` function in GCTA (v. 1.94.0)¹⁹. We then estimated the narrow-sense *cis*-heritability ($cis-h^2_{SNP}$) of each gene using a linear mixed model (LMM) accounting for the same covariates used in the *cis*-eQTL mapping procedure, with the variance components of the model being estimated using restricted maximum likelihood (REML) via the `--greml` function in GCTA²⁰. We then conducted a likelihood ratio test (LRT) comparing the model with the additive GRM versus the null model of no genetic component and corrected resulting *P*-values using the BH-FDR procedure. We defined genes with an $P_{adj.}$ value < 0.05 as being significantly heritable. Using the same *cis*-SNPs for each significantly heritable gene, we then estimated the dominance GRM using the `--make-grm-d` function in GCTA¹⁷. We then approximated the broad sense *cis*-heritability (H^2_{SNP}) of each gene (i.e., the proportion of phenotypic variance explained by both dominant (δ^2_{SNP}) and additive (h^2_{SNP}) genetic effects) by fitting an additive and dominant (AD) model using REML. We then used the LRT to evaluate whether the addition of the dominance component significantly improved the proportion of variation explained in the expression level of the gene and corrected resulting *P*-values using the BH-FDR procedure. For genes with a $P_{adj.}$ value < 0.05 , dominance variance was considered to account for a significant proportion of the *cis* heritability in expression levels.

Across all genes tested in *cis* in the AAG cohort, and for which a GRM could be made for ($n = 14,563$), we observed that the median $cis-h^2_{SNP}$ was 0.165 (**Fig. R1a**). We noted a significant difference in the median $cis-h^2$ estimates of genes with expression levels considered significantly heritable (LRT $P_{adj.} < 0.05$) ($n = 6,757$) (median $cis-h^2 = 0.412$) to those that were not significantly heritable ($n = 7,806$; median $cis-h^2_{SNP} = 0.04$) (Wilcoxon rank-sum test; $P < 2.2 \times 10^{-16}$) (**Fig. R1b**). For the 6,757 significantly heritable genes, we were able to generate dominance GRMs for 5,863 genes. For these genes, we estimated the median H^2_{SNP} to be 0.474, the median $cis-h^2_{SNP}$ to be 0.4311 and the median $cis-\delta^2_{SNP}$ to be 6×10^{-6} with this difference between the two latter categories being highly statistically significant (Wilcoxon rank-sum test; $P < 2.2 \times 10^{-16}$) (**Fig. R1c**). Out of the 5,863 genes tested in this analysis, we observed two genes for which the dominance component significantly contributed to the *cis* heritability of gene expression. These two genes were *LANCL1* (LRT $P_{adj.} = 0.01$; $cis-h^2_{SNP} = 0.314$; $cis-\delta^2_{SNP} = 0.489$) and the unannotated *ENSBTAG00000031825* gene (LRT $P_{adj.} = 0.03$; $cis-h^2_{SNP} = 0.914$; $cis-\delta^2_{SNP} = 0.057$).

Fig. R1 (Supplementary Fig. S11 in revised manuscript): *Cis*-heritability of gene expression is predominantly governed by additive genetic variance. (a) Distribution of narrow-sense *cis*-heritability ($cis-h^2$) estimates for all genes ($n = 14,563$) tested in the *cis*-eQTL analysis in the all animal group (AAG) cohort and for which a GRM could be constructed. (b) Distribution of $cis-h^2$ values for genes not considered significantly heritable (Non-sig; likelihood ratio test (LRT) $P_{adj.} > 0.05$) ($n = 7,806$) and for those considered significantly heritable (Sig.; LRT $P_{adj.} < 0.05$) ($n = 6,757$) (c) Distribution of broad-sense heritability (H^2_{SNP}) estimates for 5,863 significantly heritable genes with this estimate decomposed into the proportion explained by additive variance (light green) and dominance variance (δ^2_{SNP} ; orange). The box plots in **a**, **b** and **c** cover the interquartile range with the median line denoted at the centre and mean being denoted by the black square box. The whiskers extend to the most extreme data point that is no more than $1.5 \times$ IQR from the edge of the box. P -values in panels **b** and **c** are inferred from the Wilcoxon rank-sum test between the heritability estimates within each group.

2. Evaluating addition of the non-additive component in linear regression analysis.

We also tested for the presence of a non-additive effect across all independent *cis*-eQTLs in each group separately using a likelihood ratio test (LRT). Briefly, for all independent *cis*-eQTLs in the AAG, bTB- and bTB+ groups, using the `lm()` function in R, we fitted an additive linear model between expression levels and genotype at an independent *cis*-eQTL locus with genotypes coded as 0, 1 and 2 depending on the number of alternative allele copies, while accounting for the same covariates used in the *cis*-eQTL analysis using TensorQTL. Separately, we also fitted the same linear model that also included a non-additive component whereby heterozygous animals were coded as 1 and homozygous reference

and homozygous alternative animals were coded as 0 (*Im_dom*). We then performed a likelihood ratio test (LRT) between the nested *Im_add* model and the complex *Im_dom* model using the *lrtest* function from the *lmtree* R package to determine whether adding the non-additive component significantly improved the model fit. The LRT statistic was compared to a chi-squared distribution with 1 degree of freedom and resulting *P*-values were corrected for multiple testing using the BH-FDR procedure. Complex models encompassing a non-additive component that achieved an *P_{adj.}* value < 0.05 provided evidence that the effect of a *cis*-eQTL on the expression level of a gene deviated significantly from additivity.

The results of this analysis are shown in **Table R2**: Across 2235, 3736 and 11,092 independent *cis*-eQTL effects tested for evidence of dominance in the bTB+, bTB- and AAG cohorts, we identified a total of three, four and 31 independent *cis*-eQTLs displaying deviating significantly from additivity. This equates to 0.13%, 0.11% and 0.28% of all tested *cis*-eQTLs, respectively. Moreover, with respect to the comments related to **Fig. 4e** and **Fig. 4d**, we tested these *cis*-eVariants for evidence of an effect that deviated from additivity in both the bTB+ and bTB- cohorts using this approach. In the bTB- cohort, we did not observed any nominally significant evidence of deviation from additivity for these variants (variant = Chr29:50294904:G:A, gene = *IFITM3*, $LRT_{bTB-} P = 0.601$ (**Fig. 4e**) and variant = Chr26:40304855:C:A, gene = *RGS10*, $LRT_{bTB-} P = 0.599$ (**Fig. 4g**). Similarly, with respect to the bTB+ group, we did not observe any nominally significant evidence that adding a non-additive component would improve the proportion of variation explained by the model (variant = Chr29:50294904:G:A, gene = *IFITM3*, $LRT_{bTB+} P = 0.426$ (**Fig. 4e**) and variant = Chr26:40304855:C:A, gene = *RGS10*, $LRT_{bTB+} P = 0.112$ (**Fig. 4g**).

Table R2: Summary of independent *cis*-eQTLs (FDR *P_{adj.}* < 0.05) identified across the reactor (bTB+), control (bTB) and combined all animal group (AAG) cohorts. The table displays the total number of independent *cis*-eQTLs tested with a dominance component, along with the number and percentage of genes displaying significant (FDR *P_{adj.}* < 0.05) evidence of a non-additive effect based on the likelihood ratio test comparing a nested additive model to a complex model including both additive and non-additive components.

	bTB+ (n = 60)	bTB- (n = 63)	AAG (n = 123)
Total number of independent cis -eQTLs	2,497	4,238	12,165
Cis -eQTLs tested for a non-additive effect	2235	3736	11,092
Number of cis -eQTLs displaying significant evidence ($LRT P_{adj.} < 0.05$) of a deviation from additivity	3	4	31
Percentage (%) of cis -eQTLs tested displaying significant evidence deviation from additivity	0.13	0.11	0.28

Bearing these results in mind, we have therefore added the following paragraphs into the Results section of the revised manuscript.

L259 - L260: Mapping *cis*-expression quantitative trait loci and estimating the contribution of non-additive effects

L327 - L352: Across all genes tested in the *cis*-eQTL mapping analysis for the AAG cohort, we constructed an additive genomic relationship matrix (GRM) for $n = 14,563$ of these genes and observed that the median *cis* narrow-sense heritability of gene expression ($cis-h^2_{SNP}$) was 0.165 (**Supplementary Fig. S11a**). We noted a significant difference in the median $cis-h^2_{SNP}$ estimates of genes with expression levels considered significantly heritable (LRT $P_{adj.} < 0.05$) ($n = 6,757$; median $cis-h^2_{SNP} = 0.412$) to those that were not significantly heritable ($n = 7,806$; median $cis-h^2_{SNP} = 0.04$) (Wilcoxon rank-sum test; $P < 2.2 \times 10^{-16}$) (**Supplementary Fig. S11b**). For the 6,757 genes with significantly heritable expression levels, we were able to construct dominance GRMs for 5,863 (86.7%) of these genes. For these genes, we estimated the median *cis* broad sense heritability (H^2_{SNP}) of gene expression to be 0.474, the median $cis-h^2_{SNP}$ to be 0.431 and the median *cis* heritability of dominance effects ($cis-\delta^2_{SNP}$) to be 6×10^{-6} with the difference between the two latter categories being statistically significant (Wilcoxon rank-sum test; $P < 2.2 \times 10^{-16}$) (**Supplementary Fig. S11c**). Out of the 5,863 genes tested in this analysis, we observed two genes (0.03%) for which the dominance component contributed a significant proportion to the $cis-H^2_{SNP}$ of gene expression. These two genes were *LANCL1* (LRT $P_{adj.} = 0.01$; $cis-h^2_{SNP} = 0.314$; $cis-\delta^2_{SNP} = 0.489$) and the unannotated *ENSBTAG00000031825* gene (LRT $P_{adj.} = 0.03$; $cis-h^2_{SNP} = 0.914$; $cis-\delta^2_{SNP} = 0.057$).

We also tested to infer what proportion of *cis*-eQTLs mapped using the linear model could be better explained by the addition of a non-additive (dominance) component. Of the 2,497, 4,238 and 12,165 independent *cis*-eQTLs identified in the bTB+, bTB- and AAG cohorts, respectively, we fitted a non-additive component in the linear model for 2,235, 3,736 and 11,092 *cis*-eQTLs, as these *cis*-eQTLs had ≥ 1 animal in both homozygous groups. Of these *cis*-eQTLs tested for evidence of non-additive effects in the bTB+, bTB- and AAG cohorts, we identified a total of three, four and 31 independent *cis*-eQTLs displaying significant evidence of a deviation from additivity, which equated to 0.13%, 0.11% and 0.28% of all tested *cis*-eQTLs in each group, respectively. Overall, these results indicate that dominance variance makes a negligible contribution to the *cis*-heritability of gene expression and mapping of *cis*-eQTLs in cattle is robustly supported by an additive model.

We have also added in the following sections into the Methods section:

L879 – L913:

Estimating the *cis*-SNP based heritability of gene expression

To quantify the contribution of additive and dominance effects to the *cis*-SNP based heritability of gene expression, for each normalised gene tested in the *cis*-eQTL analysis of the AAG cohort, we extracted the SNPs ± 1 Mb away from the TSS of the associated gene and constructed an additive genomic relationship matrix (GRM) using these SNPs via the `--make-grm` function in GCTA (v. 1.94.0)¹⁹. We then estimated the narrow-sense *cis*-heritability ($cis-h^2_{SNP}$) for each gene using a linear mixed model (LMM) accounting for the same covariates used in the *cis*-eQTL mapping procedure, with the variance components of the model being estimated using restricted maximum likelihood (REML) via the `--greml` function in GCTA²⁰. We then conducted a likelihood ratio test (LRT) comparing the model with the additive GRM versus the null model of no genetic component and corrected resulting *P*-values using the BH-FDR procedure. We defined genes with a $P_{adj.}$ value < 0.05 as being significantly heritable in this analysis. Following this, using the same *cis*-SNPs for each significantly heritable gene, we then calculated the dominance GRM using the `--make-grm-d` function in GCTA¹⁷. We then approximated the broad sense *cis*-heritability (H^2_{SNP}) for each gene (i.e., the proportion of phenotypic variance explained by both dominant (δ^2_{SNP}) and additive (h^2_{SNP}) genetic effects) by fitting an additive and

dominant (AD) model using REML. We then used the LRT to evaluate whether the addition of the dominance component significantly improved the proportion of variation explained in the expression level of the gene and corrected resulting P -values using the BH-FDR procedure. For genes with a $P_{\text{adj.}}$ value < 0.05 , dominance variance was considered to account for a significant proportion of the cis H^2_{SNP} in expression levels.

Testing the contribution of non-additive effects in cis -eQTL mapping

To examine what proportion of independent cis -eQTLs could be better explained by including a non-additive component in the linear model we fitted two linear models for all independent cis -eQTLs across all three groups using the $lm()$ function in R. The first model represented the same linear model used in the standard cis -eQTL mapping analysis with genotypes coded as 0, 1 and 2 depending on the number of alternative allele copies (nested). The second model was the same as the first model but also included a non-additive (dominance) component with both homozygous groups coded as 0 and heterozygotes coded as 1 (complex). For the independent cis -eQTLs for which we could fit a dominance component for, we then used the $lrtest$ function from the $lmtest$ R package v. 0.9.40 to conduct an LRT test between the complex and nested model to determine if the addition of the non-additive component in the linear model significantly improved model fit. The LRT statistic was compared to a chi-squared distribution with one degree of freedom and resulting P -values were corrected for multiple testing using the BH-FDR procedure. Complex models encompassing a non-additive component that achieved a $P_{\text{adj.}}$ value < 0.05 provided significant evidence for a deviation from additivity.

- Line 743: Why were inferred covariates preferentially retained over known covariates? Wouldn't results be more interpretable using known covariates when available?

AUTHOR: We thank Reviewer 1 for raising this issue. We agree that retaining known covariates would facilitate the interrogation of the factors that contribute to variability in gene expression. However, the reason why we decided to preferentially retain inferred covariates was because that many bovine-eQTL studies are void of rich or contain inaccurate metadata meaning that they rely heavily on this data driven approach from the transcriptomics data. Hence, if future researchers wish to implement our cis -eQTL pipeline, they may benefit from this approach that more accurately captures the factors contributing to gene expression variability.

We note, however, that in our analysis, using this approach, the covariates that were removed were RNA-sequencing batch (technical confounder) in the bTB+, bTB- and AAG cohorts, and genotype PC1 (another inferred covariate) in the bTB- cohort (**Supplementary Table S9-S11**). We would not consider assessing the effects of sequencing batch biologically relevant and we have previously discussed the contribution of genotype PC1 in the AAG as a confounder in the differential expression analysis as in this cohort of animals, it largely represents the proportion of Holstein Friesian ancestry (**Fig. 2b**, **Supplementary Fig. S6a**).

- Line 500: The authors outline a cell deconvolution analysis for which all the data are already available. Why was this analysis not performed and relevant cell proportions explored as covariates in models?

We thank Reviewer 1 for raising this point. We had considered computationally deconvolving our bulk-RNA-seq data to identify differences in cell type proportions between the bTB+ and bTB- cohorts and to map cell-type ieQTLs. There are two main issues that prevented us from conducting this analysis.

Firstly, while publicly available bovine single-cell RNA-seq (scRNA-seq) data from peripheral blood (PB) (the tissue analysed here) exists, there is currently no scRNA-seq data from PB of cattle infected with *M. bovis*. This creates challenges in terms of inferring the cell type proportions in the bTB+ group as using a signature reference matrix derived from healthy animals introduces biological bias and reduces overall deconvolution performance²¹. We note that scRNA-seq data of peripheral blood mononucleated cells (PBMCs) from cattle infected with *M. tuberculosis* is available (GSE218065) albeit not published in the literature. However, this would create challenges in terms of robustly deconvoluting our bTB+ group as *M. tuberculosis* elicits an attenuated immune response compared to *M. bovis*, which is reflected in the bovine peripheral blood transcriptome²² and PBMCs differ significantly from whole blood in terms of its immune gene expression profile²³.

Future work involving collaboration with the FarmGTEx Consortium to generate PB scRNA-seq data of cattle infected with *M. bovis* can address this issue. This will subsequently allow the use of the above-mentioned integrative genomic techniques leveraging the data analysed in this study.

We have included the following modification to the discussion section of the manuscript

L546 – L550: Once appropriate PB single-cell RNA-seq data from cattle infected with *M. bovis* becomes available, computational deconvolution²⁴ of this bulk RNA-seq data would enable the characterisation of cell type differences between these two groups and attribute the gene expression changes to an increase or decrease in the abundance of a particular cell type. In this regard, the Cattle Cell Atlas²⁵ will be a significant resource for this endeavour.

- Line 602-622: This paragraph begins with relevant discussion of the limitations of the TWAS approach. However, the authors then discuss several non-significant gene findings, hypothesizing that these genes did not meet significance criteria due to low statistical power. Just as possible, given a larger sample size, these genes would be confirmed as nonsignificant. There is no evidence in the current to support findings regarding these genes and interest in “immunologically relevant genes” is not at all specific to the few genes highlighted here. The authors “cherry picking” appears to be a priori interests that do not align with the genome-wide approaches used.

We thank Reviewer 1 for raising this point. We agree with Reviewer 1 and accept that it is inappropriate to speculate whether genes found not to be significant after the permutation procedure in our TWAS analysis would become significant if a larger sample size was present in the reference set. We have decided to remove our commentary of these genes from the text and modified the section to read as follows:

L676 - L682: However, the permutation scheme itself is highly conservative and as such, true causal genes associated with the trait of interest may be filtered out owing to insufficient power²⁶. Conversely, as animals in the bTB+ reference panel have a confirmed bTB diagnosis and are maintained for bTB diagnostics potency testing, it is possible that some of the results from the TWAS may be confounded by horizontal pleiotropy owing to the same causal variant having independent effects on both expression and the trait²⁶.

Minor comments

- Table 1: The distance column does not capture unique information per row. Thus, this information should be included in the Table title and this column removed.

AUTHOR: Removed and information included in the table title.

- Figure S11: I believe loesse should be loess.

AUTHOR: Corrected. We also italicised the word *cis* in the figure title.

- Line 368-380: It is a bit difficult to follow the use of “upregulated” vs “downregulated” here. Suggest referring to DEGs as bTB- or bTB+ associated to clarify.

AUTHOR: Corrected. We have removed the use of the word “upregulated”

- Figure 5: Please verify that the cow icon color matches associated data. Based on the legend, I think that the control (blue) icon is on infected data.

AUTHOR: Corrected. The cow icons were correct, but the legend was incorrect. We have now fixed this issue. We also modified the axis label of panel **b** and **c** to italicise the “Z” in Z-score

- Line 507-511: This text is not necessary as it is stating a well-established statistical principle i.e. that sample size and effect size impact statistical power.

AUTHOR: We have removed the part of the sentence which reads “and our power of detection was dependent on sample size, which has been previously reported”

Reviewer #2 (Remarks to the Author):

This is an extensive study and generally well written.

AUTHOR: The authors thank Reviewer 2 for taking the time to review our manuscript and for providing positive feedback and comments.

Some comments:

Line 143-154: Move method description to methods section and retain only relevant results here.

AUTHOR: Done. We believe what is retained represents relevant results only.

Lines 150-154: We do know that many interactions occur at the biological level due to the influence of both internal and external factors. One of these factors could be responsible for the conflicting phenotypes by the different methods for these 3 individual animals so retaining them in the analyses could affect the analyses outcome. Instead, why not remove and try to understand the reason behind the conflicting results?

AUTHOR: We thank Reviewer 2 for raising this point.

All the bTB- and bTB+ animals received a negative and positive test result respectively based on the single intradermal comparative tuberculin skin test (SICTT) that was conducted by a licenced veterinary practitioner. The standard interpretation of the SICTT to yield a positive result is if the induration at the injection site of purified protein derivative-bovine (PPDb) is positive (≥ 4 mm) and exceeds the induration at the injection site of PPD-avian (PPDa) by more than 4mm. It is this test that is the basis

for our classification of an animal as bTB⁻ or bTB⁺. Notwithstanding the fact that the bTB⁻ animals originate from herds with no recent history of bTB disease, the SICTT possesses a specificity of >99% under Irish field conditions²⁷. Specificity in this instance refers to the percentage of non-infected animals that the SICTT correctly identifies as not having bTB. In contrast to the SICTT, the ancillary interferon-gamma (IFN- γ) release assay (IGRA) test is less specific, with estimates ranging from 90.8–96.6% depending on the purified protein derivative (PPD)-bovine (PPDb) IFN- γ value minus the PPD-avian (PPDa) IFN- γ value (Δ PPD) cut-off used²⁸. It has been reported at an animal level in Great Britain (GB) (which has similar production systems to Ireland), that the specificity of the SICTT is 99.98% based on standard interpretation^{29,30}. Therefore, one false positive result is expected approximately every 5582 animals (1 *minus* the specificity). Given these findings Goodchild, et al.²⁹ estimated that the positive predictive value (the probability that an animal testing positive is truly infected) of the SICTT across GB was 91.8% indicating that if an animal tests positive on the SICTT, there is a high probability that they are truly infected with *M. bovis*.

The sensitivity of the IGRA test is estimated to be 88%³¹ although this can vary depending on the Δ PPD cut off used³². Sensitivity in this instance refers to the ability of the IGRA test to correctly classify an infected animal as positive. Since 2018, the threshold used by the Department of Agriculture Food and the Marine (DAFM) in Ireland for determining a positive result in the IGRA test is a Δ PPD value ≥ 80 (**Supplementary Table S2, Supplementary Figure S1**). This has replaced the previous threshold of Δ PPD ≥ 0 for determining a positive result. In recent work by Madden and colleagues³³ animals that would have been classified as positive using the old testing regime but that remain in the herd as a consequence of the new testing regime (as they are classified IFN- γ negative), are at increased likelihood of a positive bTB diagnosis during follow-up testing when compared with other test-negative animals.

Bearing all these factors in mind, with respect to the bTB⁻ negative animal C050 that elicited a positive reaction to the IFN- γ test (Δ PPD = 263.1), it would be incorrect to classify this animal as bTB⁺ due to the specificity of the IGRA being lower, meaning that classifying this animal as bTB⁺ would likely be a false positive designation. With respect to the two bTB⁺ animals that elicited a negative result based on the IGRA test (T007, Δ PPD = 36.0; T062, Δ PPD = -52.3), given that they both have a positive SICTT result, there is already a high probability that they are infected with *M. bovis*. For T007, based on the old testing thresholds (Δ PPD ≥ 0) this animal would have been classified as IFN- γ positive, and considering the findings by Madden, et al.³³ discussed above, there would have been an increased likelihood that this animal would have obtained a positive bTB diagnosis in follow-up testing if classified as bTB⁻ based on the IGRA test alone. For the animal T062, given the sensitivity estimates of the IGRA test, as we are testing $n = 60$ confirmed SICTT positive animals, we would expect approximately six bTB⁺ animals to be classified as IFN- γ negative, so it is not surprising that we are observing a negative result for this animal.

We agree with Reviewer 2 that subsetting these animals and examining their transcriptomes more closely might uncover biological factors that contribute to their misclassification based on the IGRA test that may potentially contribute to improved bTB diagnostics. However, that is beyond the scope of this work. The primary purpose of this work was to determine the genetic architecture underpinning the peripheral blood transcriptomic response during bTB disease.

Nonetheless, we have modified the manuscript to in the Results and the Methods section to clearly specify our designation of the animals as bTB⁻ and bTB⁺ respectively based on the SICTT.

L149 – L151: Based on their original SICTT results, these animals were still designated as bTB⁻ and bTB⁺, respectively and included in subsequent analyses.

L712 – L715: For the purposes of this study, the experimental designation of an animal as bTB– or bTB+ was based on their SICTT result. This is because the SICTT is more specific than the IGRA test and the positive predictive value (i.e., the probability that an animal testing positive from the SICTT is truly infected) is estimated to be 91.8%²⁹.

Figure 1: There was no mention of IFN- γ assay in the methods section.

AUTHOR: We have modified the Methods section to include a statement about of the IFN- γ optical density measurements were generated for the animals.

L699 – L704: As an ancillary diagnostic test carried out in series, all animals were tested for *M. bovis* infection using the whole blood IFN- γ release assay (IGRA) test (BoviGAM[®] – Prionics AG, Switzerland) following the procedure described by Clegg, et al.³². The criterion for IFN- γ test positivity was a test result difference greater than 80 ELISA units for the purified protein derivative (PPD)-bovine (PPDb) IFN- γ value minus the PPD-avian (PPDa) IFN- γ value (Δ PPD)³².

Figure 1:SNP array data- Why did you not consider combining this with culled SNPs from the RNA-Seq data of the same animals? This strategy would make available more population specific SNPs and a better quality imputed variant set for downstream analysis. See Koyama et al., 2024. While most SNP arrays do present the opportunity to capture SNPs spaced at defined distances of the genome, they may not capture SNPs present in all genes nor population specific variants.

Koyama, S., Liu, X., Koike, Y. et al. Population-specific putative causal variants shape quantitative traits. *Nat Genet* 56, 2027–2035 (2024). <https://doi.org/10.1038/s41588-024-01913-5>

AUTHOR: We thank Reviewer 2 for raising this point and for sharing the recently published paper by Koyama et al., (2024). We agree that SNP arrays do not capture all population-specific genomic variation. However, we believe that bovine SNP arrays are biased towards and capture common European taurine breed variation effectively (the population studied here) but are poor at capturing African/indicine variation³⁴. Nonetheless, Riggio, et al.³⁴ showed that imputation performance from a high-density (HD) array (e.g. the ~600K Affymetrix Axiom[™] Genome-Wide BOS-1 array used here) using the Dutta, et al.³⁵ Global Reference WGS Panel (leveraged here) was similar between African/Indicine and European/Taurine cattle with lower performance observed for Asian indicine cattle (not studied here) as these animals are less represented on the HD bovine arrays. This indicates that genomic variation in our European taurine study cohort, that is predominantly Holstein-Friesian based on comprehensive pedigree data and integrative analysis of this data with genotype information (see **Fig. 2B**, **Supplementary Table S1** and **Supplementary Fig. S6A**) are captured well in the first instance by the Affymetrix Axiom[™] Genome-Wide BOS-1 array and secondly, by being well represented in the WGS Global Reference Panel³⁵.

We had not considered performing this type of analysis due to the points made above. Moreover, given the extensive linkage disequilibrium (LD) present in European taurine cattle breeds³⁶, we believe that common haplotypes can be imputed with a high degree of accuracy into the reference panel. Additionally, given that the variants that generally impute less accurately, i.e. rare variants (minor allele frequency (MAF) < 0.05) are removed prior to any downstream analyses, we would not expect breed-specific variants of “rarer” European taurine breeds (e.g. Charolais or Belgian Blue) to be retained that would significantly impact our results.

However, we did further explore this hypothesis. To do this, we followed the PigGTE_x pipeline (https://github.com/FarmGTE_x/PigGTE_x-Pipeline-v0/blob/master/02_RNA-

Seq/03_SNP_calling.smk)³⁷ and used GATK (v. 4.3.0.0)³⁸ to call variants directly from the RNA-seq data. We filtered out low-quality SNPs using the filtering options; FS > 30.0, QD < 2.0 and DP < 4.0. We then merged each individual vcf file together to generate an RNA-seq cohort-based vcf file. We then removed SNPs with a relaxed call rate < 0.8 and restricted our analysis to biallelic SNPs on bovine autosomes. From this analysis, a total of 37,856 SNPs were present in the RNA-seq variant call cohort, of which 2,059 (5.4%) were shared with the variants genotyped on the SNP array. Following merging of the RNA-seq variant calls with the remapped high-density array genotypes, a total of 589,165 SNPs were present, an increase of 35,797 (6.4%) on the 553,368 SNPs present in the initial analysis prior to imputation.

Following the imputation pipeline leveraged in the initial analysis, we imputed the combined array-genotyped and RNA-seq called variants up to WGS, first phasing using Beagle and then imputing using Minimac4. We observed that imputing the combined set of array-genotyped and RNA-seq called (DNA+RNA) variants had negligible impact on imputation quality across all allele frequency bins (**Figure R2**). In total, after filtering rare and poor-quality imputed SNPs using the same methodology described during the original analysis (MAF < 0.05, $R^2 < 0.6$, HWE $P < 0.0001$ and SNP call rate < 0.95) a total of 3,874,170 SNPs remained, an increase of 7,664 (0.198%) compared to the 3,866,506 that were originally retained in the AAG cohort (DNA). Additionally, we identified 16,725 SNPs that were private to the DNA+RNA data set. We calculated the LD (genotype squared correlation (r^2)) between these variants and variants residing ± 1 Mb away that were present in both the filtered DNA+RNA and the DNA data sets. We observed that all the 16,725 private RNA-seq variants were in complete LD ($r^2 = 1$) with at least one other variant previously retained in the original imputation analysis. This result indicates that imputing variants from RNA-seq data in this study cohort, while identifying more SNPs, has not captured population-specific effects or additional information that would warrant replication of the study.

Fig. R2: Imputation performance (R^2) of all imputed variants when imputing from high-density (HD) array genotype data alone (DNA; green) versus a combined set of HD-array genotyped variants and variants called from RNA-seq data (DNA+RNA; orange).

Lines 224-227: It is not clear why only a subset of DEG was used for overrepresentation analysis.

AUTHOR: We thank Reviewer 2 for raising this point. The reason for using a subset of DEGs ($FDR P_{adj.} < 0.01$) was because in previous work by our group, we encountered issues associated using many DEGs as input for Ingenuity Pathway Analysis (IPA®) and therefore decided to reduce the number of DEGs used as input for the enrichment analysis.

However, we have repeated our g:Profiler analysis using the entire set of 2,592 significant DEGs ($FDR P_{adj.} < 0.05$) and from this analysis, we identified a total of 287 significantly impacted pathways ($FDR P_{adj.} < 0.05$) (**Fig. R3**) of which 205 (71%) were present in the initial analysis. Of the 82 pathways identified in this analysis that were not identified in the initial analysis, many encompassed non-specific terms including the *aerobic respiration* and *negative regulation of mRNA processing* gene-ontology biological process (GO-BP) terms and *Cristae formation* Reactome pathway. However, this set also included terms related to those identified in the initial analysis including *regulation of defence response to virus* and *regulation of response to type II interferon* GO terms, that are consistent with results we initially reported. Overall, we are confident that using either all DEGs or a subset of highly significant DEGs produces similar results, which indicate that the genes perturbed during bTB disease in peripheral blood play a role in innate immune response and host-pathogen interaction pathways. To avoid confusion and to enable reproducibility, we have decided not to include this updated analysis in the revised version of the manuscript.

Fig. R3: Functional enrichment results of all significantly ($FDR P_{adj.} < 0.05$) differentially expressed genes used as input. Jitter plot of significantly impacted pathways/ Gene Ontology (GO) terms identified across the Gene

Ontology (GO) Biological Processes (GO:BP), Reactome (REAC) and Kyoto Encyclopaedia of Genes and Genomes (KEGG) databases using g:Profiler. The data points are coloured according to the corresponding database.

To inform the reader, of our reason to select only the highly significant DE genes, we have included the following statement in the Methods section.

L969 – L971: Additionally, to minimise the computational burden associated with IPA[®], as input for the overrepresentation analysis of DEGs, we selected a subset of DEGs with an FDR $P_{adj.}$ value < 0.01.

Lines 363- 384: Results indicate that only few variants were found associated with the expression of some genes. I think these results could have been different if the SNPs generated by genotyping with a chip were those culled from the RNA-Seq data, or a combination of both strategies followed by imputation, thereby increasing the proportion of population specific variants. See line 553 and comment about this in the discussion section.

AUTHOR: We thank Reviewer 2 for raising this point. We believe that Reviewer 2 is referring to our analysis testing the hypothesis that *cis*-eGenes regulated by bTB+ magnified *cis*-eVariants or *cis*-eGenes regulated by bTB- magnified *cis*-eVariants or both are overrepresented in the 2,592 differentially expressed genes (DEGs). Our results indicated that these gene sets were not overrepresented in our DEGs (chi-square test; $P > 0.5$ for all tests). We believe that this is a positive outcome, in terms of our differential expression analysis (DEA), as it suggests that our significant DEGs are not being driven by different genetic structure between the groups, as illustrated in **Fig. 2B**, **Supplementary Table S1**, and **Supplementary Fig S6B**. This outcome is to be expected given that we controlled for such variation in the DEA analysis by incorporating the genotype PCs.

Regarding imputing a combined target set of high-density (HD) genotype data and variants called from RNA-seq data, we showed above (**Figure R2**) that performing such an analysis had no discernible impact on imputation performance. Additionally, it is likely that breed specific variants for “rarer” breeds in our cohort (Charolais, Belgian Blue (**Supplementary Table S1**)) will be filtered out as we are only considering variants with a MAF > 0.05 given our sample size of $n = 123$ in the analysis of the all-animal group (AAG) cohort.

Bearing this factor in mind, we have modified the results section slightly to provide some interpretation to our results of not observing that our *cis*-eGenes were overrepresented in our DEGs.

L407 - L409: These results suggest that the DEGs are not being inflated by different genetic structures in the bTB+ and bTB- groups (**Fig. 2a**), which is to be expected given that we controlled for such genetic structures in the DEA.

Line 445: Table 2?

AUTHOR: We thank Reviewer 2 for identifying this typographical error. In the revised manuscript, the table number (3) is correct.

Line 646: Were all the studied animals sacrificed in this study? If yes, provide further details.

AUTHOR: We thank the reviewer for raising this point. We note that we have full ethical approval for this study as detailed in the *Ethics approval and consent to participate* section of the manuscript. Yes,

all the bTB+ animals were sacrificed as part of this study. To reflect this, we have modified the methods section to include the following statement.

L698 – L716: The *M. bovis*-infected cattle (bTB+) were selected from a panel of naturally infected animals maintained for on-going tuberculosis surveillance at the Department of Agriculture, Food, and the Marine (DAFM) Backweston Laboratory Campus farm (Celbridge, Co. Kildare, Ireland). These animals were skin tested by an experienced veterinary practitioner and had positive single intradermal comparative tuberculin test (SICTT) results where the skin-fold thickness response to purified protein derivative (PPD)-bovine (PPDb) exceeded that of PPD-avian (PPDa) by at least 12 mm. As an ancillary diagnostic test carried out in series, all animals were tested for *M. bovis* infection using the whole blood IFN- γ release assay (IGRA) test (BoviGAM[®] – Prionics AG, Switzerland) following the procedure described by Clegg, et al.³². The criterion for IFN- γ test positivity was a test result difference greater than 80 ELISA units for the purified protein derivative (PPD)-bovine (PPDb) IFN- γ value minus the PPD-avian (PPDa) IFN- γ value (Δ PPD)³². Skin testing and blood sampling was conducted under statutory regulations according to the European Union (EU) Council trading Directive 64/432/EEC. Following the study, the bTB+ animals were removed by DAFM to a licensed commercial abattoir where they were slaughtered humanely as per national and EU regulations. Carcasses were examined for TB-like lesions by DAFM Veterinary Inspectors and recorded on national databases. During *post-mortem* examination, all the animals disclosed multiple lesions consistent with bovine tuberculosis. The non-infected control animals (bTB-) were selected from bTB-free cattle herds (all SICTT negative) and with no recent history of *M. bovis* infection and were not sacrificed as part of this study.

Line 659: Consider revising as- Genomic DNA (supplementary note 1) from all animals were genotyped...

AUTHOR: Done.

Line 816: Specify the three groups for clarity to readers and explain the rationale for the AAG group (were appropriate) since the aim of the study is finding factors explaining the most variance between the bTB- and bTB+ groups.

AUTHOR: Done. We have added in the three codes for all the groups (bTB-, bTB+, and AAG) at the beginning of this Methods section.

We also included the following statement at the beginning of the “*cis*-eQTL mapping” subsection of the Methods to explain our rationale for assessing each group separately (bTB- and bTB+) and for mapping *cis*-eQTLs in the merged group (AAG).

L8799 – L801: We mapped *cis*-eQTLs in each group (bTB- and bTB+) separately to identify context-specific *cis*-eVariants, and we mapped *cis*-eQTLs in the merged AAG group to maximise our power and to characterise interaction *cis*-eQTLs (ieQTLs) associated with bTB status.

Discussion: Comment on the application of the findings vis-a-vis the objectives of the Cattle FarmGTEx project.).

AUTHOR: Done. We have added in the following statement at the end of the first paragraph of the Discussion section

L516 – L521: Additionally, as our data set represents one of the few *in-vivo cis*-eQTL mapping analyses conducted for cattle with an infectious disease³⁹, our eQTL and TWAS results align with and support the Cattle GTEx Consortium’s objectives of: 1) mapping context-specific *cis*-eQTLs across different conditions; 2) contributing to the precision breeding of disease (in this case bTB) resistant animals; and 3) serving as a reference for veterinary medicine and comparative genomics, particularly in relation to bTB under a One Health framework.

References

- 1 Umans, B. D., Battle, A. & Gilad, Y. Where Are the Disease-Associated eQTLs? *Trends Genet* **37**, 109-124 (2021). <https://doi.org/10.1016/j.tig.2020.08.009>
- 2 Barreiro, L. B. *et al.* Deciphering the genetic architecture of variation in the immune response to Mycobacterium tuberculosis infection. *Proc Natl Acad Sci U S A* **109**, 1204-1209 (2012). <https://doi.org/10.1073/pnas.1115761109>
- 3 Kasela, S. *et al.* Interaction molecular QTL mapping discovers cellular and environmental modifiers of genetic regulatory effects. *Am. J. Hum. Genet.* **111**, 133-149 (2024). <https://doi.org/10.1016/j.ajhg.2023.11.013>
- 4 Burnham, K. L. *et al.* eQTLs identify regulatory networks and drivers of variation in the individual response to sepsis. *Cell Genom* **4**, 100587 (2024). <https://doi.org/10.1016/j.xgen.2024.100587>
- 5 Davis, J. R. *et al.* An Efficient Multiple-Testing Adjustment for eQTL Studies that Accounts for Linkage Disequilibrium between Variants. *Am. J. Hum. Genet.* **98**, 216-224 (2016). <https://doi.org/10.1016/j.ajhg.2015.11.021>
- 6 Yao, C. *et al.* Sex- and age-interacting eQTLs in human complex diseases. *Hum. Mol. Genet.* **23**, 1947-1956 (2014). <https://doi.org/10.1093/hmg/ddt582>
- 7 Taylor-Weiner, A. *et al.* Scaling computational genomics to millions of individuals with GPUs. *Genome Biol.* **20**, 228 (2019). <https://doi.org/10.1186/s13059-019-1836-7>
- 8 Gu, H. *et al.* PCBP2 maintains antiviral signaling homeostasis by regulating cGAS enzymatic activity via antagonizing its condensation. *Nat. Commun.* **13**, 1564 (2022). <https://doi.org/10.1038/s41467-022-29266-9>
- 9 Sun, L., Wu, J., Du, F., Chen, X. & Chen, Z. J. Cyclic GMP-AMP synthase is a cytosolic DNA sensor that activates the type I interferon pathway. *Science* **339**, 786-791 (2013). <https://doi.org/10.1126/science.1232458>
- 10 Wu, J. *et al.* Cyclic GMP-AMP is an endogenous second messenger in innate immune signaling by cytosolic DNA. *Science* **339**, 826-830 (2013). <https://doi.org/10.1126/science.1229963>
- 11 Wassermann, R. *et al.* Mycobacterium tuberculosis Differentially Activates cGAS- and Inflammasome-Dependent Intracellular Immune Responses through ESX-1. *Cell Host Microbe* **17**, 799-810 (2015). <https://doi.org/10.1016/j.chom.2015.05.003>
- 12 Zhu, Z. *et al.* Integration of summary data from GWAS and eQTL studies predicts complex trait gene targets. *Nat. Genet.* **48**, 481-487 (2016). <https://doi.org/10.1038/ng.3538>
- 13 Ring, S. C. *et al.* Variance components for bovine tuberculosis infection and multi-breed genome-wide association analysis using imputed whole genome sequence data. *PLoS One* **14**, e0212067 (2019). <https://doi.org/10.1371/journal.pone.0212067>
- 14 Giambartolomei, C. *et al.* Bayesian test for colocalisation between pairs of genetic association studies using summary statistics. *PLoS Genet.* **10**, e1004383 (2014). <https://doi.org/10.1371/journal.pgen.1004383>
- 15 Powell, J. E. *et al.* Congruence of additive and non-additive effects on gene expression estimated from pedigree and SNP data. *PLoS Genet.* **9**, e1003502 (2013). <https://doi.org/10.1371/journal.pgen.1003502>

- 16 Cui, L. *et al.* Dominance is common in mammals and is associated with trans-acting gene expression and alternative splicing. *Genome Biol.* **24**, 215 (2023). <https://doi.org/10.1186/s13059-023-03060-2>
- 17 Zhu, Z. *et al.* Dominance genetic variation contributes little to the missing heritability for human complex traits. *Am. J. Hum. Genet.* **96**, 377-385 (2015). <https://doi.org/10.1016/j.ajhg.2015.01.001>
- 18 Palmer, D. S. *et al.* Analysis of genetic dominance in the UK Biobank. *Science* **379**, 1341-1348 (2023). <https://doi.org/10.1126/science.abn8455>
- 19 Yang, J., Lee, S. H., Goddard, M. E. & Visscher, P. M. GCTA: a tool for genome-wide complex trait analysis. *Am J Hum Genet* **88**, 76-82 (2011). <https://doi.org/10.1016/j.ajhg.2010.11.011>
- 20 Yang, J. *et al.* Common SNPs explain a large proportion of the heritability for human height. *Nat. Genet.* **42**, 565-569 (2010). <https://doi.org/10.1038/ng.608>
- 21 Vallania, F. *et al.* Leveraging heterogeneity across multiple datasets increases cell-mixture deconvolution accuracy and reduces biological and technical biases. *Nat. Commun.* **9**, 4735 (2018). <https://doi.org/10.1038/s41467-018-07242-6>
- 22 Villarreal-Ramos, B. *et al.* Experimental infection of cattle with Mycobacterium tuberculosis isolates shows the attenuation of the human tubercle bacillus for cattle. *Sci. Rep.* **8**, 894 (2018). <https://doi.org/10.1038/s41598-017-18575-5>
- 23 van der Sijde, F. *et al.* RNA from stabilized whole blood enables more comprehensive immune gene expression profiling compared to RNA from peripheral blood mononuclear cells. *PLoS ONE* **15**, e0235413 (2020). <https://doi.org/10.1371/journal.pone.0235413>
- 24 Newman, A. M. *et al.* Robust enumeration of cell subsets from tissue expression profiles. *Nat Methods* **12**, 453-457 (2015). <https://doi.org/10.1038/nmeth.3337>
- 25 Fang, L. *et al.* Cattle Cell Atlas: a multi-tissue single cell expression repository for advanced bovine genomics and comparative biology. (2024). <https://doi.org/10.21203/rs.3.rs-4631710/v1>
- 26 Gusev, A. *et al.* Integrative approaches for large-scale transcriptome-wide association studies. *Nat Genet* **48**, 245-252 (2016). <https://doi.org/10.1038/ng.3506>
- 27 Clegg, T. A. *et al.* Using latent class analysis to estimate the test characteristics of the γ -interferon test, the single intradermal comparative tuberculin test and a multiplex immunoassay under Irish conditions. *Vet Microbiol* **151**, 68-76 (2011). <https://doi.org/10.1016/j.vetmic.2011.02.027>
- 28 de la Rua-Domenech, R. *et al.* Ante mortem diagnosis of tuberculosis in cattle: a review of the tuberculin tests, gamma-interferon assay and other ancillary diagnostic techniques. *Res Vet Sci* **81**, 190-210 (2006). <https://doi.org/10.1016/j.rvsc.2005.11.005>
- 29 Goodchild, A. V., Downs, S. H., Upton, P., Wood, J. L. & de la Rua-Domenech, R. Specificity of the comparative skin test for bovine tuberculosis in Great Britain. *Vet Rec* **177**, 258 (2015). <https://doi.org/10.1136/vr.102961>
- 30 Gormley, E. *et al.* Identification of risk factors associated with disclosure of false positive bovine tuberculosis reactors using the gamma-interferon (IFN γ) assay. *Vet Res* **44**, 117 (2013). <https://doi.org/10.1186/1297-9716-44-117>
- 31 Gormley, E., Doyle, M. B., Fitzsimons, T., McGill, K. & Collins, J. D. Diagnosis of Mycobacterium bovis infection in cattle by use of the gamma-interferon (Bovigam) assay. *Vet Microbiol* **112**, 171-179 (2006). <https://doi.org/10.1016/j.vetmic.2005.11.029>
- 32 Clegg, T. A., Doyle, M., Ryan, E., More, S. J. & Gormley, E. Characteristics of Mycobacterium bovis infected herds tested with the interferon-gamma assay. *Prev Vet Med* **168**, 52-59 (2019). <https://doi.org/10.1016/j.prevetmed.2019.04.004>
- 33 Madden, J. M. *et al.* The impact of changing the cut-off threshold of the interferon-gamma (IFN- γ) assay for diagnosing bovine tuberculosis in Ireland. *Prev Vet Med* **224**, 106129 (2024). <https://doi.org/10.1016/j.prevetmed.2024.106129>

- 34 Riggio, V. *et al.* Assessment of genotyping array performance for genome-wide association studies and imputation in African cattle. *Genet Sel Evol* **54**, 58 (2022). <https://doi.org/10.1186/s12711-022-00751-5>
- 35 Dutta, P. *et al.* Whole genome analysis of water buffalo and global cattle breeds highlights convergent signatures of domestication. *Nat Commun* **11**, 4739 (2020). <https://doi.org/10.1038/s41467-020-18550-1>
- 36 de Roos, A. P., Hayes, B. J., Spelman, R. J. & Goddard, M. E. Linkage disequilibrium and persistence of phase in Holstein-Friesian, Jersey and Angus cattle. *Genetics* **179**, 1503-1512 (2008). <https://doi.org/10.1534/genetics.107.084301>
- 37 Teng, J. *et al.* A compendium of genetic regulatory effects across pig tissues. *Nat Genet* **56**, 112-123 (2024). <https://doi.org/10.1038/s41588-023-01585-7>
- 38 McKenna, A. *et al.* The Genome Analysis Toolkit: a MapReduce framework for analyzing next-generation DNA sequencing data. *Genome Res.* **20**, 1297-1303 (2010). <https://doi.org/10.1101/gr.107524.110>
- 39 Li, J. *et al.* Applying multi-omics data to study the genetic background of bovine respiratory disease infection in feedlot crossbred cattle. *Front Genet* **13**, 1046192 (2022). <https://doi.org/10.3389/fgene.2022.1046192>

Reviewer #1 (Remarks to the Author):

I appreciate the extensive analytical work undertaken by the authors to address my questions and concerns. I have no further requests.

AUTHOR: We thank Reviewer 1 for taking the time to review our manuscript.

Reviewer #2 (Remarks to the Author):

My comments have been well handled and I do not have further comments.

AUTHOR: We thank Reviewer 2 for taking the time to review our manuscript.